SciPost Physics

Submission

# On the Boundary Conformal Field Theory Approach to Symmetry-Resolved Entanglement

Giuseppe Di Giulio[1,2], René Meyer[1,2], Christian Northe[3,1,2], Henri Scheppach[1,2], Suting Zhao[1,2]

**1** Institute for Theoretical Physics and Astrophysics, Julius Maximilian University Würzburg, Am Hubland, 97074 Würzburg, Germany
**2** Würzburg-Dresden Cluster of Excellence on Complexity and Topology in Quantum Matter ct.qmat
**3** Department of Physics, Ben-Gurion University of the Negev, David Ben Gurion Boulevard 1, Be'er Sheva 84105, Israel
* christian.northe@physik.uni-wuerzburg.de

December 31, 2022

## Abstract

We study the symmetry resolution of the entanglement entropy of an interval in two-dimensional conformal field theories (CFTs), by relating the bipartition to the geometry of an annulus with conformal boundary conditions. In the presence of extended symmetries such as Kac-Moody type current algebrae, symmetry resolution is possible only if the boundary conditions on the annulus preserve part of the symmetry group, i.e. if the factorization map associated with the spatial bipartition is compatible with the symmetry in question. The partition function of the boundary CFT (BCFT) is then decomposed in terms of the characters of the irreducible representations of the symmetry group preserved by the boundary conditions. We demonstrate that this decomposition already provides the symmetry resolution of the entanglement spectrum of the corresponding bipartition. Considering the various terms of the partition function associated with the same representation, or charge sector, the symmetry-resolved Rényi entropies can be derived to all orders in the UV cutoff expansion without the need to compute the charged moments. We apply this idea to the theory of a free massless boson with $U(1)$, $\mathbb{R}$ and $\mathbb{Z}_2$ symmetry.

# 1   Introduction

Since the early days of quantum mechanics entanglement has been considered one of the crucial and most interesting properties of quantum systems [1]. In the last two decades, a renewed interest in this subject has lead to many insights into various branches of physics, ranging from quantum gravity and holography to critical and topological many-body systems [2–6]. Quantifying the entanglement between a spatial region $A$ in a given system and its complement $B$ is particularly important. We assume that the Hilbert space $\mathcal{H}$ of the entire system factorizes as $\mathcal{H} = \mathcal{H}_A \otimes \mathcal{H}_B$, where $\mathcal{H}_A$ encodes the degrees of freedom in the region $A$ and $\mathcal{H}_B$ the ones in $B$. Given a pure state $|\psi\rangle \in \mathcal{H}$, the reduced density matrix of $A$ is then given by $\rho_A \equiv \mathrm{tr}_B |\psi\rangle\langle\psi|$, where $\mathrm{tr}_B$ is the trace over the Hilbert space $\mathcal{H}_B$. The Rényi entropies, defined as

$$S_n = \frac{1}{1-n} \ln \mathrm{tr}\rho_A^n, \tag{1}$$

quantify the bipartite entanglement, where $n$ is integer. Upon analytic continuation to complex values of $n$, the limit $n \to 1$ of the Rényi entropies yields the entanglement entropy

$$S_1 = -\mathrm{tr}\left(\rho_A \ln \rho_A\right). \tag{2}$$

For convenience, in this text we refer to both entanglement entropy (2) and Rényi entropies (1) simply as entanglement entropies. Entanglement and its measures proved to be extremely useful in the study of critical systems, which in the continuum limit can be described by conformal field theories (CFT). In a 1+1-dimensional CFT with central charge $c$, when the entire system on an infinite line is in the ground state and $A$ is an interval of length $\ell$, the entanglement entropies obey the following behaviour [7–10]

$$S_1 = \frac{c}{3} \log \frac{\ell}{\epsilon}, \qquad S_n = \frac{c}{6}\frac{n+1}{n} \log \frac{\ell}{\epsilon}, \tag{3}$$

at leading order in the UV cutoff $\epsilon \ll \ell$. This result and the others corresponding to different states where $A$ is still a single interval can be retrieved by mapping the geometry of interest to an annulus and exploiting boundary conformal field theory (BCFT) techniques [11–13]. An important feature of this approach is the possibility to access the entanglement spectrum of the theory upon determining the conformal dimensions of the operators in the BCFT. This is usually not achieved through other methods, as for instance the twist fields method, which only provide the moments of the reduced density matrix.

Recently, sparked by experimental results [14–17] and the developments of new theoretical tools [18–20], a growing interest in the interplay between entanglement and symmetries emerged. Given a system with a global symmetry and a spatial bipartition as described above, the amount of entanglement in the different charge sectors can be quantified by the symmetry-resolved entanglement entropies. The symmetry-resolved entanglement entropies have been computed in 2D CFTs [19–28], integrable and free quantum field theories [29–36], as well as lattice models [23, 26, 37–52]. Moreover, the symmetry resolution of other entanglement quantifiers, such as negativity [53–57],

relative entropies and distances [58, 59] and operator entanglement [60, 61] has been studied in the CFT setup as well. The symmetry-resolved entanglement entropies have been also considered in the context of the AdS/CFT correspondence and computed in some examples [62–66], finding the expected matching between bulk and boundary results.

Despite the number of results, a better understanding of the symmetry resolution of entanglement is still desirable. It will be interesting to understand better the conditions necessary for the equipartition of entanglement namely the fact that, at leading order in the cutoff expansion, the symmetry-resolved entanglement entropies are independent of the charge sector. Computing the symmetry-resolved entanglement entropies requires in principle the knowledge of the entanglement spectrum resolved in the various charge sectors, which is a formidable task from the analytic point of view. This problem is bypassed by first computing partition functions on Riemann surfaces with flux, known as charged moments, which then lead to the symmetry-resolved entropies through a Fourier transform. Studying theories where the exact resolution of the entanglement spectrum is known can help in obtaining new insights into the relation between entanglement and symmetries. Moreover, this would allow accessing the symmetry-resolved entanglement entropies without first computing the charged moments, c.f. the examples in [40, 41], where the analytic knowledge of the resolution of the entanglement spectrum in XXZ spin chains was used to achieve this.

The goal of this paper is to advance the understanding of symmetry-resolved entanglement entropy in the BCFT setup. The BCFT description is useful, since it provides access to the full spectrum of the subregion density matrix [20]. This not only allows for the computation of the charged moments but also the direct computation of the charged partition function, and therefore, as we show below, for an easier, more efficient computation of the symmetry-resolved entanglement entropy. Indeed, in presence of a symmetry, the BCFT partition function decomposes into contributions from the various charge sectors. The symmetry-resolved entanglement entropy in a given charge sector can be directly computed from the terms in the BCFT partition function associated with that charge. This was first noted in the appendix of [22]. Furthermore, the BCFT picture allows for a regularization more in the spirit of quantum field theory [12]. Instead of putting the theory on the lattice, the BCFT prescription maps the entangling interval to an annulus with boundary conditions on both ends (see figure 1). Indeed, since fields are distribution valued, spacetime boundaries need the specification of boundary conditions.

We apply the general BCFT formalism to the compact and non-compact free boson CFTs. Calculations of the *total* entanglement of the compact boson from this perspective are found in [67–69]. We resolve entanglement with respect to the symmetry groups $U(1)$, $\mathbb{R}$ and $\mathbb{Z}_2$. Particular attention is directed to the fact that the boundary conditions need to not only preserve Virasoro symmetry but also the additional symmetry with respect to which we wish to resolve. In the case of the free compact boson, this additional symmetry group is $U(1)$, in the case of the non-compact free boson $\mathbb{R}$. Since $\mathbb{Z}_2$ is a subgroup of both $U(1)$ and $\mathbb{R}$, it is also a symmetry of the free boson CFT. For the CFTs we consider, certain boundary conditions indeed break the $U(1)$ or $\mathbb{R}$ symmetry, respectively, and therefore resolution with respect to these groups is no longer possible. In these cases we show, using the BCFT approach outlined above, that symmetry resolution with respect to a remnant $\mathbb{Z}_2$ is still possible. In general, this new approach provides access to all higher order terms in the UV cutoff expansion of symmetry-resolved Rényi entropies. We expand on the analysis of [20] by the consideration of the non-compact case, the $\mathbb{Z}_2$ symmetry and the incorporation of the higher order terms in the symmetry-resolved entanglement entropy. The results for $U(1)$ and $\mathbb{R}$ resolutions we report here are obtained in two ways. The main results are calculated directly from the BCFT partition functions. They are cross-checked by computing the charged moments and their Fourier transform.

The paper is structured as follows. In section 2 we give a general introduction to symmetry-resolved entanglement entropy and define the charged moments and charged partition functions. In section 3 we review the BCFT setup for entanglement entropy

and review the factorization mapping given in [12]. We introduce the boundary states of the compact free boson CFT and give the partition functions for different boundary conditions. We discuss the decompactification limit to obtain the partition functions for the non-compact free boson. From the partition functions we calculate the entanglement entropies and show that this approach recovers the universal term in the lowest order of the system size. In section 4, we present our calculation of the symmetry-resolved entanglement entropy directly from the BCFT spectrum. We report the symmetry-resolved entanglement entropy for the compact and non-compact free boson. We check these calculations with a more conservative approach, where we calculate the charged moments and their Fourier transform using the BCFT approach. We also provide a resolution with respect to $\mathbb{Z}_2$ from the exact knowledge of the spectrum. This resolution is also possible for boundary conditions which break the $U(1)$ and $\mathbb{R}$ symmetries. We provide a summary of our results and further directions for research in section 5. Furthermore, we provide two appendices, appendix A and appendix B, in which we clarify and extend certain aspects of the analysis in the paper.

## 2 Symmetry-Resolved Entanglement

Consider a quantum system endowed with a global abelian symmetry group $G$, generated by the charge operator $\mathcal{Q}$. Under the assumption that the charge is local, we can decompose it into the contribution in the subsystem $A$ and the one in its complement, namely $\mathcal{Q} = \mathcal{Q}_A \otimes \mathbf{1}_B + \mathbf{1}_A \otimes \mathcal{Q}_B$, where $\mathbf{1}_i$ is the identity in the Hilbert space $\mathcal{H}_i$, $i = A, B$. We are interested in the cases where the system is in a pure state $|\psi\rangle$, which is an eigenstate of $\mathcal{Q}$. When this happens, we have $[|\psi\rangle\langle\psi|, \mathcal{Q}] = 0$ and, tracing this commutator over $\mathcal{H}_B$, it follows that $[\rho_A, \mathcal{Q}_A] = 0$, where $\rho_A$ is the reduced density matrix of $A$. The last identity implies that $\rho_A$ has a block-diagonal structure, where each block corresponds to an eigenvalue $Q$ of the charge operator $\mathcal{Q}_A$. It reads

$$\rho_A = \bigoplus_Q \Pi_Q \rho_A = \bigoplus_Q P_A(Q) \rho_A(Q), \tag{4}$$

where $\Pi_Q$ is the projector onto the eigenspace associated to $Q$ and $P_A(Q) \equiv \operatorname{tr}(\Pi_Q \rho_A)$ is the probability of having $Q$ as outcome of a measurement of $\mathcal{Q}_A$. Notice that, since $G$ is an abelian group, the eigenvalues $Q$ label the irreducible representations of the group itself.

The decomposition (4) ensures the normalization $\operatorname{tr}\rho_A(Q) = 1$ for any value of $Q$ and therefore one can quantify the amount of entanglement in the sector with charge $Q$ via the symmetry-resolved Rényi entropies

$$S_n(Q) = \frac{1}{1-n} \ln \operatorname{tr}\rho_A(Q)^n, \tag{5}$$

and the symmetry-resolved entanglement entropy

$$S_1(Q) = -\operatorname{tr}\left[\rho_A(Q) \ln \rho_A(Q)\right]. \tag{6}$$

The block-diagonal structure in (4) allows to decompose the total entanglement entropy as

$$S_1 = \sum_Q P_A(Q) S_1(Q) - \sum_Q P_A(Q) \ln P_A(Q) \equiv S_{1,\mathrm{c}} + S_{1,\mathrm{f}}. \tag{7}$$

The first summand in (7) is known as configurational entanglement entropy [14, 70–72], while the second one as fluctuation (or number) entanglement entropy [14, 73–75]. They encode information about the entanglement within each charge sector and the fluctuations between different sectors, respectively. Notice that a decomposition similar to (7) does not hold in general for the Rényi entropies. However, we observe that, under certain assumptions, it is still possible to identify a configurational and

a fluctuational contribution to $S_n$. In order to do so, let us plug (4) into (1) and, exploiting the fact that the trace of a block diagonal matrix is the sum of the traces of the individual blocks, we obtain

$$S_n = \frac{1}{1-n} \ln \left[ \sum_Q [P_A(Q)]^n \operatorname{tr}\rho_A(Q)^n \right]. \tag{8}$$

In the most general case, the logarithm in (8) does not split into the sum of logarithms and therefore configurational and a fluctuational contributions cannot be identified. However, we can assume equipartition, i.e. that $\operatorname{tr}\rho_A(Q)^n$ does not depend on the charge $Q$. Then, after defining $R_n \equiv \operatorname{tr}\rho_A(Q)^n$, we can rewrite (8) as

$$S_n = \frac{1}{1-n} \ln R_n + \frac{1}{1-n} \log \left[ \sum_Q [P_A(Q)]^n \right] \equiv S_{n,\mathrm{c}} + S_{n,\mathrm{f}}. \tag{9}$$

By taking the limit $n \to 1$ of $S_{n,\mathrm{c}}$ and $S_{n,\mathrm{f}}$ we obtain, within the restricted case we are considering, $S_{1,\mathrm{c}}$ and $S_{1,\mathrm{f}}$ respectively. For this reason, one can interpret $S_{n,\mathrm{c}}$ and $S_{n,\mathrm{f}}$ as configurational and fluctuation Rényi entropies, but only when $\operatorname{tr}\rho_A(Q)^n$ does not depend on $Q$. Notice that, when this happens, the system is characterised by an *exact* equipartition of the entanglement, namely (5) and (6) do not depend on $Q$ at any order. In section 4, we discuss an instance where $\operatorname{tr}\rho_A(Q)^n$ is actually independent of the charge sector and therefore the decomposition (9) is valid.

The key object to compute is the replica partition function $\mathcal{Z}_n(Q)$ at fixed charge $Q$,

$$\mathcal{Z}_n(Q) = \operatorname{tr}(\Pi_Q \rho_A^n), \tag{10}$$

which allows to write the symmetry-resolved entanglement entropies as

$$S_n(Q) = \frac{1}{1-n} \log \frac{\mathcal{Z}_n(Q)}{\mathcal{Z}_1(Q)^n}, \qquad S_1(Q) = \lim_{n \to 1} S_n(Q). \tag{11}$$

The computation of $\mathcal{Z}_n(Q)$ requires the knowledge of the entanglement spectrum and its symmetry resolution. This information is often difficult to access, in particular through analytic techniques. A possible way to overcome this problem relies on suitably re-expressing the projector $\Pi_Q$. More explicitly, let us consider two particular cases, namely $G = U(1)$, $G = \mathbb{R}$ and $G = \mathbb{Z}_N$. When $G = U(1)$ or $G = \mathbb{R}$, we can exploit a Fourier representation of the projector $\Pi_Q$ and $\mathcal{Z}_n(Q)$ can be written as [19, 20]

$$\mathcal{Z}_n(Q) = \frac{1}{2\pi} \int_{-\lambda}^{\lambda} d\mu \, e^{-\mathrm{i}\mu Q} \operatorname{tr}\left( e^{\mathrm{i}\mu \mathcal{Q}_A} \rho_A^n \right). \tag{12}$$

When the charges $Q$ are discrete ($U(1)$ group), the integration bound is $\lambda = \pi$, whereas for continuous charges $Q$ ($\mathbb{R}$ group), the integration bound tends to infinity, $\lambda \to \infty$. We observe that $\mathcal{Z}_n(Q)$ is the Fourier transform of the charged moments

$$\mathcal{Z}_n(\mu) = \operatorname{tr}\left( e^{\mathrm{i}\mu \mathcal{Q}_A} \rho_A^n \right). \tag{13}$$

When $G = \mathbb{Z}_N$, the restricted charge operator has $N$ eigenvalues. The projector $\Pi_Q$ can be expanded over the $N$ elements of the group and $\mathcal{Z}_n(Q)$ reads [19]

$$\mathcal{Z}_n(Q) = \frac{1}{N} \sum_{j=0}^{N-1} e^{-\frac{2\pi \mathrm{i} j Q}{N}} \operatorname{tr}\left( e^{\frac{2\pi \mathrm{i} j \mathcal{Q}_A}{N}} \rho_A^n \right), \tag{14}$$

which leads to the definition of the corresponding charged moments

$$\mathcal{Z}_n(j) = \operatorname{tr}\left( e^{\frac{2\pi \mathrm{i} j \mathcal{Q}_A}{N}} \rho_A^n \right), \qquad \mathcal{Z}_n(Q) = \frac{1}{N} \sum_{j=0}^{N-1} e^{-\frac{2\pi \mathrm{i} j Q}{N}} \mathcal{Z}_n(j). \tag{15}$$

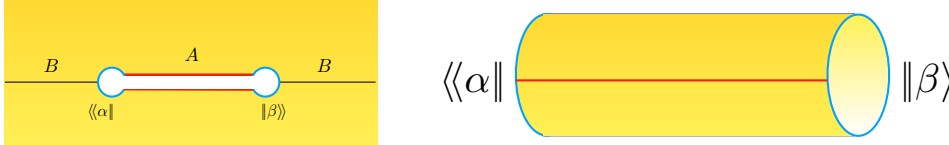

Figure 1: The BCFT setup of the entanglement entropy. Small disks of raidus $\epsilon$ are excised around the entangling point (left panel). The resulting manifold is mapped into an annulus (right panel) by a conformal transformation in such a way that the small disks encircling the entangling points become the boundaries of the annulus (blue circles).

In section 4.2, we work out the case of $\mathbb{Z}_2$ as explicit example.

By first computing the charged moments and then $\mathcal{Z}_n(Q)$ using (12) and (14), the symmetry-resolved entanglement entropies have been successfully computed in various theories and for different entangling regions [19, 20, 29, 37, 63–65]. This is usually done exploiting the fact that the charged moments can be seen as partition functions of QFTs defined on an $n$-sheeted Riemann surface pierced by an Aharanov-Bohm flux [19]. As we will explain in detail in section 3, the BCFT approach for the computation of the entanglement entropies is particularly useful for the purpose of the symmetry resolution, as it gives directly access to how the elements of the entanglement spectrum are distributed in the various symmetry sectors. Thus, one can compute $\mathcal{Z}_n(Q)$ and the symmetry-resolved entanglement entropies without resorting to the charged moments and, moreover, to all orders in the UV cutoff. In section 3, we expand on this idea and we provide various examples involving free CFTs.

## 3    Entanglement Entropy and BCFT

It has been observed in [12] that the standard procedure of decomposing Hilbert space $\mathcal{H} = \mathcal{H}_A \otimes \mathcal{H}_B$ where $A, B$ are spatial regions is a priori incomplete when working with quantum fields. Quantum fields are distributions and need to be smeared over regions of space, and thus a spatial domain cannot simply be cut sharply into separate spatial regions $A, B$ without indicating how the fields should behave at the boundaries between $A$ and $B$. In short boundary conditions need to be specified at $\partial A$ and $\partial B$.

When $A$ is a single entangling interval this is achieved as pictorially represented in figure 1. First, small disks[1] of radius $\epsilon$, which takes on the role of UV cutoff, are excised around the two endpoints of $A$. Second, boundary conditions $\alpha$ and $\beta$ are imposed on these cutoff disks. Since $A$ and $B$ share their boundaries, the Hilbert space for $B$ is also characterized by the boundary conditions $\alpha$ and $\beta$. This procedure is encoded in a factorization map [12]

$$\iota : \mathcal{H} \to \mathcal{H}_{A,\alpha\beta} \otimes \mathcal{H}_{B,\alpha\beta}, \qquad \iota : |\psi\rangle \mapsto \iota |\psi\rangle, \tag{16}$$

for $|\psi\rangle \in \mathcal{H}$. The boundary conditions thus depend on the particular choice of factorization $\iota$. The reduced Hilbert space $\mathcal{H}_{A,\alpha\beta}$ and its reduced density matrices are obtained by tracing over $\mathcal{H}_{B,\alpha\beta}$,

$$\rho_A = \mathrm{Tr}_{\mathcal{H}_{B,\alpha\beta}} \left[ \iota |\psi\rangle \langle\psi| \iota^\dagger \right]. \tag{17}$$

Our interest lies on CFT and we choose boundary conditions $\alpha, \beta$ which preserve conformal symmetry, i.e. $T = \bar{T}|_{boundary}$ [76,77]. The new manifold with excised disks is mapped [12] into an annulus by a conformal transformation, see the right panel in figure 1. In this coordinate frame, traces of $\rho_A^n$ are readily evaluated as BCFT

---

[1]A priori one might choose any shape. However, any shape, topologically equivalent to a disk, can be mapped into a disk by a conformal transformation. Disks respect the local rotation invariance and thus they represent an optimal choice for our purposes.

partition functions. When $n \neq 1$, $n$ annuli are glued along $A$ to construct a replica annulus [12, 13] of width $W$ and circumference $2\pi n$. In the ground state of an infinite system, such as the one leading to (3), the width is $W = 2\log\left[\frac{\ell}{\epsilon} - 1\right] \approx 2\log\frac{\ell}{\epsilon} + \mathcal{O}(\epsilon)$, where $\ell$ is the length of the interval $A$; the width $W$ is discussed in [13] for various other states.

In terms of the modular nome,

$$q = e^{2\pi i \tau} = e^{-2\pi^2/W}, \qquad \tilde{q} = e^{-2\pi i/\tau} = e^{-2W}, \qquad \tau = i\pi/W, \tag{18}$$

the reduced density matrix is [13]

$$\rho_A = \frac{q^{L_0 - c/24}}{Z_{\alpha\beta}}. \tag{19}$$

Thus, one arrives at a relation between the traces of the density matrix and BCFT partition functions

$$\mathcal{Z}_n = \mathrm{Tr}_{\alpha\beta}\left[\rho_A^n\right] = \frac{Z_{\alpha\beta}(q^n)}{(Z_{\alpha\beta}(q))^n}, \tag{20}$$

where we abbreviated the symbol for the trace $\mathrm{Tr}_{\mathcal{H}_{A,\alpha\beta}} \to \mathrm{Tr}_{\alpha\beta}$ and used the standard expression for a BCFT partition function, $Z_{\alpha\beta}(q) = \mathrm{Tr}_{\alpha\beta}\left[q^{L_0 - \frac{c}{24}}\right]$ with the Virasoro zero mode $L_0$ and the central charge $c$. The denominator ensures $\mathrm{Tr}_{\alpha\beta}[\rho_A] = 1$.

In this work, we consider CFTs with extended symmetries, namely with a symmetry algebra larger than the Virasoro algebra. A BCFT has in general less symmetry than the original CFT since the introduction of a boundary breaks some symmetries and so the mapping (16) breaks these as well. The remaining symmetry algebra, after imposing $\iota$, is called $\mathcal{A}$ in the following. In these cases, the Hilbert space decomposes,

$$\mathcal{H}_{A,\alpha\beta} \equiv \mathcal{H}_{\alpha\beta} = \bigoplus_i \mathcal{H}_i^{\oplus n_{\alpha\beta}^i}, \tag{21}$$

where $i$ runs over the allowed representations of the extended symmetry algebra $\mathcal{A}$. Since we require the boundary conditions to preserve conformal symmetry, the extended algebra contains the Virasoro algebra, $\mathsf{Vir} \subset \mathcal{A}$. The multiplicities $n_{\alpha\beta}^i$ are determined by the imposed boundary conditions $\alpha$ and $\beta$. The BCFT partition function decomposes then into characters $\chi_i(q) = \mathrm{Tr}_{\mathcal{H}_i}\left[q^{L_0 - \frac{c}{24}}\right]$ for representations $i$,

$$Z_{\alpha\beta}(q^n) = \sum_i n_{\alpha\beta}^i \chi_i(q^n) = \langle\!\langle \alpha \| \tilde{q}^{\frac{1}{n}\left(L_0 - \frac{c}{24}\right)} \| \beta \rangle\!\rangle. \tag{22}$$

In the last equality, the BCFT partition functions are computed in the $S$-dual channel via boundary states

$$\|\alpha\rangle\!\rangle = \sum_j B_\alpha^j |j\rangle\!\rangle, \tag{23}$$

where $|j\rangle\!\rangle$, satisfying $(L_n - \bar{L}_{-n})|j\rangle\!\rangle = 0$, is an Ishibashi state for the $j^{\text{th}}$ representation of $\mathcal{A}$. An example is given below for the free boson. They satisfy an "orthogonality relation" $\langle\!\langle j | \tilde{q}^{L_0 - \frac{c}{24}} | i \rangle\!\rangle = \chi_i(\tilde{q})\delta_{ij}$. The coefficients $B_\alpha^j$ stand in relation with the $n_{\alpha\beta}^i$ via the Cardy constraint and the reader is referred to [13, 78] for details.

## 3.1 Free Boson BCFT

The ideas of the previous subsection are now exemplified in the free boson CFT. This section recapitulates results which already exist in the literature, see for instance [67].

The free boson CFT on the plane is governed by the action

$$S = \frac{g}{2}\int_\Sigma \mathrm{d}\tau \mathrm{d}\sigma (\partial_\mu \varphi)(\partial^\mu \varphi) = g\int_\Sigma \mathrm{d}^2 z (\partial\varphi)(\bar\partial\varphi). \tag{24}$$

This action is invariant under conformal transformations and also under $U(1)$ transformations implemented by $\varphi + const$. In fact, the symmetry algebra of the massless

free boson is given by a $\hat{\mathfrak{u}}(1) \times \hat{\mathfrak{u}}(1)$ Kac-Moody algebra and two copies of the Virasoro algebra, generated by the currents

$$J = \sum_{n\in\mathbb{Z}} a_n z^{-n-1}, \quad T = \sum_{n\in\mathbb{Z}} L_n z^{-n-2}, \qquad \bar{J} = \sum_{n\in\mathbb{Z}} \bar{a}_n \bar{z}^{-n-1}, \quad \bar{T} = \sum_{n\in\mathbb{Z}} \bar{L}_n \bar{z}^{-n-2}.$$

(25)

The $U(1)$ symmetry responsible for the shift $\varphi + const.$ is generated by the $U(1)$ charge $a_0$. Additionally, there are the spectrum generating modes $a_n$ with $n \neq 0$. Together they form the $\hat{\mathfrak{u}}(1)$ algebra. The modes satisfy the algebra

$$[a_n, a_m] = n\delta_{n+m,0}, \tag{26a}$$

$$[L_n, a_m] = -na_{n+m}, \tag{26b}$$

$$[L_n, L_m] = (n-m)L_{n+m} + \frac{c}{12}n(n^2-1)\delta_{n+m,0}, \tag{26c}$$

and similarly for $\bar{a}_n$ and $\bar{L}_n$. The spectrum of the compact boson with periodicity $\varphi \simeq \varphi + 2\pi R$ is given by primary states $|(m,w)\rangle$ and their descendants, where $m \in \mathbb{Z}$ is a momentum quantum number and $w \in \mathbb{Z}$ is one for the winding sectors. The primaries have conformal weights and $\hat{\mathfrak{u}}(1)$ charges

$$h_{m,w} = \frac{Q_{m,w}^2}{8\pi g}, \qquad\qquad Q_{m,w} = \frac{m}{R} - \frac{4\pi g\, wR}{2}, \tag{27a}$$

$$\bar{h}_{m,w} = \frac{\overline{Q}_{m,w}^2}{8\pi g}, \qquad\qquad \overline{Q}_{m,w} = \frac{m}{R} + \frac{4\pi g\, wR}{2}. \tag{27b}$$

BCFTs are defined on Riemann surfaces with a boundary, usually taken to be the upper half plane or an annulus. Two types of boundary conditions may be imposed on these boundaries, which preserve a single copy of $\hat{\mathfrak{u}}(1)$, defined by the gluing conditions $J = \pm\bar{J}|_{bdy}$. Hence, in the notation of the previous section, $\mathcal{A} = \hat{\mathfrak{u}}(1)$.

For $J = \bar{J}|_{bdy}$ the gluing conditions result in Neumann (N) boundary conditions

$$\partial_\sigma \varphi|_{bdy} = 0, \tag{28}$$

and for $J = -\bar{J}|_{bdy}$ they result in Dirichlet (D) boundary conditions

$$\partial_\tau \varphi|_{bdy} = 0. \tag{29}$$

In the D case, the value of the bosonic field may take different values, $\varphi_0$ and $\varphi_0'$, at the two boundaries of the annulus, see figure 1. In the case of N boundary conditions, the structure is identical when describing the model in terms of the dual bosonic field $\theta$ [78]. If the boson $\varphi$ is decomposed into left- and right-movers, $\varphi = \phi + \bar{\phi}$, then the dual field is decomposed as $\theta = \phi - \bar{\phi}$. In the case of N boundaries, the dual field $\theta$ assumes fixed values $\theta_0$ and $\theta_0'$ to either end of the annulus.

Each boundary condition is encoded in a boundary state

$$\|N(\theta_0)\rangle\rangle = \sqrt{\sqrt{\pi g}R} \sum_{w\in\mathbb{Z}} e^{2\pi i g\, Rw\,\theta_0} |(0,w)\rangle\rangle_N, \tag{30a}$$

$$\|D(\varphi_0)\rangle\rangle = \frac{1}{\sqrt{\sqrt{4\pi g}R}} \sum_{m\in\mathbb{Z}} e^{i\frac{m}{R}\varphi_0} |(m,0)\rangle\rangle_D, \tag{30b}$$

with Ishibashi states built on top of the bulk primaries (27),

$$N: \quad |(0,w)\rangle\rangle_N = \exp\left(-\sum_{n=1}^{\infty} \frac{1}{n} a_{-n}\bar{a}_{-n}\right) |(0,w)\rangle, \tag{31a}$$

$$D: \quad |(m,0)\rangle\rangle_D = \exp\left(\sum_{n=1}^{\infty} \frac{1}{n} a_{-n}\bar{a}_{-n}\right) |(m,0)\rangle. \tag{31b}$$

The entropy of the boundary fields is given by Affleck-Ludwig g-factors [79], which in the model at hand are given by

$$\mathbf{g}_N = \langle 0 \| N(\theta_0) \rangle\!\rangle = \sqrt{R\sqrt{\pi g}}, \tag{32a}$$

$$\mathbf{g}_D = \langle 0 \| D(\phi_0) \rangle\!\rangle = \frac{1}{\sqrt{R\sqrt{4\pi g}}}. \tag{32b}$$

Note that the g-factors are independent of the parameters $\varphi_0, \theta_0$.

It is important to stress that the modes which appear in the boundary states are not the eigenstates of the entanglement hamiltonian. They are degrees of freedom present in the theory before imposing boundaries. Thus they are called bulk modes and in fact they transform under two copies of the Virasoro algebra. The eigenstates of the entanglement hamiltonian on the other hand, which we refer to as boundary fields, transform only under a single copy of the Virasoro algebra. They are the fields living at the boundary, i.e. the excised disks, and the protagonists of the BCFT. The boundary fields and bulk fields are related to each other via an $S$-dual transformation. Hence the bulk modes provide a useful means of computing the spectrum of boundary fields.

In the following we state the partition functions and spectra of all combinations of N and D conditions on the boundaries, expressed once via the boundary modes and thereafter via the bulk modes. Derivations would lead us too far afield and thus are not provided here. The reader can find a good introduction in [78].

The partition functions are computed from (22). For the case of Neumann boundaries to either end with $\theta_0$ and $\theta_0'$ it is

$$Z_{NN}(q) = \sum_{m \in \mathbb{Z}} \chi_m^{(\Delta\theta_0)}(q) = \mathbf{g}_N^2 \sum_{w \in \mathbb{Z}} e^{2\pi i g w R \Delta\theta_0} \chi_{(0,w)}(\tilde{q}), \tag{33}$$

where $\Delta\theta_0 = \theta_0 - \theta_0'$. The second and third expressions here correspond to the second and third expressions in $(22)^2$. We have introduced $\hat{u}(1)$ characters and the Dedekind $\eta$ function

$$\chi_m^{(\Delta\theta_0)}(q) = \frac{q^{h_m^{(\Delta\theta_0)}}}{\eta(q)}, \qquad \chi_{(m,w)}(q) = \frac{q^{h_{m,w}}}{\eta(q)}, \qquad \eta(q) = q^{\frac{1}{24}} \prod_{n=1}^{\infty} (1 - q^n). \tag{34}$$

The boundary fields have conformal dimensions $h_m^{(\Delta\theta_0)} = \frac{(Q_m^{(\Delta\theta_0)})^2}{8\pi g}$, where the $\hat{u}(1)$ charges lie in the NN spectrum $\sigma_{NN}$ given by

$$Q_m^{(\Delta\theta_0)} = 4\pi g \left( \frac{m}{2\pi g R} + \frac{\Delta\theta_0}{2\pi} \right), \qquad m \in \mathbb{Z}. \tag{35}$$

Comparing with (22), the first equality of (33) shows that all $\hat{u}(1)$ families with conformal weight $h_m^{(\Delta\theta_0)}$ appear for $m \in \mathbb{Z}$ with multiplicity $n_{\theta_0',\theta_0}^m = 1$. The second equality phrases the partition function in the $S$-dual channel, revealing which bulk modes run through the annulus: as can be read off from (22) and (30), these are the ones with $\hat{u}(1)$ charges $Q_{0,w}$, c.f. (27).

Similarly, for the case of DD boundary conditions with $\varphi_0$ and $\varphi_0'$ to each end of the annulus

$$Z_{DD}(q) = \sum_{w \in \mathbb{Z}} \chi_w^{(\Delta\varphi_0)}(q) = \mathbf{g}_D^2 \sum_{m \in \mathbb{Z}} e^{i\frac{m}{R}\Delta\varphi_0} \chi_{(m,0)}(\tilde{q}), \tag{36}$$

where $\Delta\varphi = \varphi_0 - \varphi_0'$. The boundary fields have conformal dimensions $h_w^{(\Delta\varphi_0)} = \frac{(Q_w^{(\Delta\varphi_0)})^2}{8\pi g}$, where the $\hat{u}(1)$ charges lie in the DD spectrum $\sigma_{DD}$ given by

$$Q_w^{(\Delta\varphi_0)} = 4\pi g \left( wR + \frac{\Delta\varphi_0}{2\pi} \right), \qquad w \in \mathbb{Z}. \tag{37}$$

---

$^2$The way to derive (33) is to take the $\| N(\theta_0) \rangle\!\rangle$ and compute their overlap $\langle\!\langle N(\theta_0') \| \tilde{q}^{L_0 - 1/24} \| N(\theta_0) \rangle\!\rangle$ exploiting that $\langle\!\langle (0, w') | \tilde{q}^{L_0 - 1/24} | (0, w) \rangle\!\rangle = \delta_{w',w} \chi_{(0,w)}(\tilde{q})$. This provides the third expression in (33). A subsequent Poisson resummation yields the second expression.

The first equality of (36) shows that all $\hat{u}(1)$ families with conformal weight $h_w^{(\Delta\varphi_0)}$ appear for $w \in \mathbb{Z}$ with multiplicity $n_{\varphi_0',\varphi_0}^w = 1$. The second equality in (36) phrases the partition function in the $S$-dual channel, revealing which bulk modes run through the annulus, namely the ones with charge $Q_{m,0}$ in (27).

The partition function for mixed boundary conditions, DN and ND, is

$$Z_{DN}(q) = Z_{ND}(q) = q^{\frac{1}{48}} \prod_{n=1}^{\infty}(1 - q^{n-\frac{1}{2}})^{-1} \quad = \mathsf{g}_N\mathsf{g}_D\left(\tilde{q}^{\frac{1}{24}}\prod_{k=1}^{\infty}(1 + \tilde{q}^k)\right)^{-1}. \quad (38)$$

Note that this partition function does not have $U(1)$ symmetry [78,80], indicated by the fact that $Z_{ND}$ does not decompose into $\hat{u}(1)$ characters $\chi_Q$ built from $a_n$ modes for $n \in \mathbb{Z}$. Instead, the spectrum is given by a twisted $\hat{u}(1)$ representation built on a primary twist field $\sigma$ of dimension $h_\sigma = 1/16$. This Fock space is constructed by modes $a_r$ with half-integral index, $r \in \mathbb{Z} + \frac{1}{2}$,

$$\mathcal{H}_{ND} = \{a_{-r_1}a_{-r_2}\ldots a_{-r_k}|\sigma\rangle\}. \quad (39)$$

Importantly, this implies the absence of the $U(1)$ charge operator $a_0$. This breaking of the $U(1)$ symmetry by mixed boundary conditions is demonstrated with standard field theoretic methods in appendix A.

Before concluding our review of the free boson BCFT, we consider the decompactification limit $R \to \infty$, so that the target space of the boson becomes $\mathbb{R}$. In the NN case the partition function becomes

$$Z_{NN}^{(\infty)} = \frac{1}{\eta(q)}\int_{\mathbb{R}}d\alpha\,q^{\frac{\alpha^2}{8\pi g}} = \mathsf{g}_{N,\infty}^2\frac{1}{\eta(\tilde{q})}, \quad (40)$$

where the g-factor is now $\mathsf{g}_{N,\infty}^2 = \sqrt{4\pi g}$. This means that *all* $\hat{u}(1)$ families occur in this BCFT with multiplicity one and conformal dimension $h_\alpha = \frac{\alpha^2}{8\pi g}$. This is in line with setting $\alpha = 2m/R$ in (35). The parameter $\theta_0$ must vanish in the $R \to \infty$ limit since the phases in (30a) would be ill-defined otherwise; $\theta_0$ is thus dropped henceforth from all N labels in the decompactification limit. Only a single $\hat{u}(1)$ family of bulk modes is required to describe this BCFT, namely that of the unit field with $h = 0$, as is indicated by the absence of any summation or integration in $\tilde{q}$ expression of (40).

In the DD case

$$Z_{DD}^{(\infty)} = \frac{q^{\frac{g}{2\pi}(\Delta\varphi_0)^2}}{\eta(q)} = \mathsf{g}_{D,\infty}^2\int_{\mathbb{R}}d\beta\,e^{i\beta\Delta\varphi_0}\frac{\tilde{q}^{h_\beta}}{\eta(\tilde{q})}, \quad (41)$$

with g-factor $\mathsf{g}_{D,\infty}^2 = \frac{1}{\sqrt{4\pi g}}$. Observe that there is only one $\hat{u}(1)$ family amongst the boundary spectrum, that of conformal dimension $h_0^{(\Delta\varphi_0)}$, c.f. (37). This is the well known observation that winding modes, $w \neq 0$, are too heavy to be present in the decompactification limit. Note that the parameter $\varphi_0$ is still non-vanishing. It is now the bulk channel in which all $\hat{u}(1)$ families appear in the annulus partition function. They have dimension $h_\beta = \frac{\beta^2}{8\pi g}$.

## 3.2 Entanglement in Free Boson Theory

Using the expressions for the partition functions in the free boson theory for NN, DD and DN boundary conditions, we now evaluate the Rényi and entanglement entropies in all cases.

We begin with the case of DD boundaries in the compact boson. Given the last expression of the spectrum in (36), and using (20), the Rényi entropies are

$$S_n^{DD} = \log \mathsf{g}_D^2 + \frac{1}{1-n}\left[\log\left(\frac{\eta(\tilde{q})^n}{\eta(\tilde{q}^{\frac{1}{n}})}\right) + \log\left(\frac{\sum_m e^{i\frac{m}{R}\Delta\varphi_0}\tilde{q}^{\frac{h_{m,0}}{n}}}{\left(\sum_k e^{i\frac{k}{R}\Delta\varphi_0}\tilde{q}^{h_{k,0}}\right)^n}\right)\right]. \quad (42)$$

The first term is the standard g-factor contribution independent of $n$ [12]. The third term accounts for the bulk CFT primaries propagating in the BCFT. The second term, accounting for all descendants in the theory, is responsible for the standard Rényi entropy. Indeed,

$$\frac{1}{1-n} \log\left(\frac{\eta(\tilde{q})^n}{\eta(\tilde{q}^{\frac{1}{n}})}\right) = \frac{W}{12}\frac{n+1}{n} + \frac{1}{1-n}\sum_{k=1}^{\infty} \log\left[\frac{(1-e^{-2Wk})^n}{1-e^{-2Wk/n}}\right], \qquad (43)$$

where $\tilde{q} = e^{-2W}$ has been used. It is straightforward to take the $n \to 1$ limit,

$$S_1^{DD} = \frac{W}{6} + \log \mathbf{g}_D^2 - \sum_{k=1}^{\infty}\left[\log(1-e^{-2Wk}) + \frac{2Wk}{1-e^{2Wk}}\right]$$

$$+ \mathbf{g}_D^2 \sum_{m\in\mathbb{Z}} e^{i\frac{m}{R}\Delta\varphi_0} \frac{\chi_{m,0}(\tilde{q})}{Z_{D(\varphi_0'),D(\varphi_0)}(q)} \log\left[-\frac{2Wh_{m,0}\,\eta(\tilde{q})Z_{D(\varphi_0'),D(\varphi_0)}(q)}{\mathbf{g}_D^2}\right]. \quad (44)$$

In later sections we illuminate the physics of $S_1^{DD}$ more clearly by rewriting it in terms of probability distributions. This analysis is repeated identically for the NN case using the last expression for the partition function (33). Thus we omit the discussion of the NN case here.

Simplifications occur for the non-compact boson. We begin with the DD case, where it is actually convenient not to use the last expression of (41) but the one in terms of $q$, which provides

$$S_n^{DD,(\infty)} = \frac{1}{1-n} \log\left[\frac{\eta^n(\tilde{q})}{\eta(\tilde{q}^{\frac{1}{n}})}\frac{\sqrt{-in\tau}}{(\sqrt{-i\tau})^n}\right]$$

$$= \frac{W}{12}\frac{n+1}{n} + \frac{1}{1-n}\sum_{k=1}^{\infty} \log\left[\frac{(1-e^{-2Wk})^n}{1-e^{-2Wk/n}}\right] - \frac{1}{2}\log\left[\frac{W}{\pi}\right] + \frac{1}{2}\frac{\log n}{1-n}. \quad (45)$$

The modular transformation $\eta(\tilde{q}) = \sqrt{-i\tau}\eta(q)$ has been used in going to the second line. Observe that a $\log W$ term arises. This might surprise readers familiar with symmetry-resolution in the free boson as this here is the regular entanglement entropy. As will be explained in detail in section 4, the $\log W$ term has to appear here. The limit $n \to 1$ is easily taken in this representation.

The NN case for the non-compact boson is in fact very simple, given the last expression in (40)

$$S_n^{NN,(\infty)} = \frac{W}{12}\frac{n+1}{n} + \frac{1}{1-n}\sum_{k=1}^{\infty} \log\left[\frac{(1-e^{-2Wk})^n}{1-e^{-2Wk/n}}\right] + \log \mathbf{g}_{N,(\infty)}^2. \qquad (46)$$

Observe the appearance of the g-factor and that it is not accompanied by an $n$-dependence.

The final case is that of mixed boundaries, ND. Regardless of a compact or non-compact target space, the partition function takes the form (38)

$$S_n^{DN} = \frac{W}{12}\frac{n+1}{n} + \frac{1}{1-n}\sum_{k=1}^{\infty} \log\left[\frac{(1+e^{-2Wk})^n}{1+e^{-2Wk/n}}\right] + \log(\mathbf{g}_D\mathbf{g}_N). \qquad (47)$$

This concludes our review of entanglement in the free boson CFT. These expressions are used in the following section to discuss the symmetry resolution in this theory in full generality.

# 4 A New Take on Symmetry Resolution

In this section we present a shift of perspective on the subject of symmetry resolution, which is possible whenever the spectrum of the entanglement BCFT is accessible. The focus is shifted away from charged moments toward the actual structure of

the entanglement spectrum (21). As will become clear, this new approach simplifies the computations of symmetry-resolved entanglement significantly and provides new structural insights into the entanglement of the free boson theory.

In order for the setup and the cutting operation described in section 3 to work for SREE, additional constraints need to be imposed. As noted in [12], the cutting operation $\iota$ does not necessarily preserve all symmetries of the system. Since the cutting operation $\iota$ is not unique, one chooses one that preserves half of the Virasoro symmetry to utilize BCFT methods. This corresponds to placing conformal boundary conditions at the entangling points.

In order to calculate the SREE for a system with some additional symmetry group, the cutting operation must also preserve said additional symmetry. This is vital, as the reduced density matrix after the cutting (16) needs to decompose as in (4). This fact can formally be expressed as follows: The condition that implies the reduced density matrix to be block-decomposed into the various charge sectors is $[\rho_A, \mathcal{Q}_A] = 0$, where both operators in the commutator are restricted to the subsystem of interest. In the language of [12], this condition can be expressed as

$$\mathrm{Tr}_{\mathcal{H}_{B,\alpha\beta}}\left(\iota[|\psi\rangle\langle\psi|, \mathcal{Q}]\iota^\dagger\right) = 0. \tag{48}$$

To check its validity, one should access the explicit form of the mapping $\iota$ defined in (16), which in general is not known. However, one can say that any $\iota$ which maps the initial CFT to a given BCFT which breaks the symmetry generated by $\mathcal{Q}$ cannot satisfy (48). In the following we mention a case where the $U(1)$ symmetry of a compact boson CFT (as well as the $\mathbb{R}$ symmetry of a non-compact boson) is broken once the theory is considered on the annulus with DN or ND boundary conditions and we argue that for this choice (48) cannot hold.

In the following we calculate the charged partition functions (10) for the compact and the non-compact free boson on the annulus with NN or DD boundaries. We consider the symmetry resolution of entanglement with respect to various symmetry groups. The results obtained are cross-checked by computing the charged moments and, in doing this, we expand on the calculation of [20] by explicitly investigating different boundary conditions. Fourier transforming the charged moments yields the $\mathcal{Z}_n(Q)$, which match the results obtained with the previous approach.

## 4.1 $U(1)$ Resolution for NN and DD Boundaries Revisited

In order to calculate the symmetry-resolved entanglement entropy, the starting point is the decomposition of the Hilbert space (21) into charge sectors in the presence of an additional symmetry $G$. The crucial observation is that the projector in the definition of the charged partition function (10) simply selects precisely one representation $\mathcal{H}_i$. The action of the free boson, eq. (24), is invariant under a translation of the angular variable, $\varphi \to \varphi + a$. This leads to the conservation of a $U(1)$ charge $\mathcal{Q}$ given by the zero mode of the current $J(z)$. Without boundaries, the symmetry is in fact $U(1) \times U(1)$. Upon implementing DD or NN boundaries, this is broken to a single $U(1)$, c.f. appendix A for a derivation. The Hilbert space of the corresponding quantum theory therefore decomposes into $\hat{\mathfrak{u}}(1)$ representations $\mathcal{H}_Q$,

$$\mathcal{H}_{\alpha\alpha} = \bigoplus_{Q \in \sigma_{\alpha\alpha}} \mathcal{H}_Q. \tag{49}$$

The charge eigenvalues $Q$ lie in the sets $\sigma_{\alpha\alpha}$ provided in (35) for NN boundaries and in (37) for DD boundaries. To improve readability, we drop the indices of the charges provided there, e.g. $Q_m^{(\Delta\theta_0)} \to Q$. The decomposition (49) is the free boson analog of (21). As explained above, ND and DN boundaries break $U(1)$ symmetry and hence do not decompose as (49). This case is discussed later.

Each representation $\mathcal{H}_Q$ consists of a $U(1)$ primary state $|Q\rangle$ with charge eigenvalue $Q$ and its descendants constructed via the modes $a_{-n}$. These representations

are irreducible[3]. The states of charge $Q$ are also Virasoro primaries with conformal dimension $h_Q = \frac{Q^2}{8\pi g}$ [82], where $g$ is the coupling introduced in the free boson action (24).

**Compact Free Boson: DD and NN boundary conditions:**  To calculate the charged partition functions (10) for the free boson, note that the spectra (33) and (36) decompose into $\hat{u}(1)$ representations. This means that the projector $\Pi_Q$ in (10) simply selects a particular $\hat{u}(1)$ character if the charge $Q$ is in the spectrum, and otherwise this expression vanishes. The probability distribution reads

$$P_A(Q) = \mathrm{Tr}_{\alpha\alpha}\left[\Pi_Q \rho_A\right] = \begin{cases} \frac{\chi_Q(q)}{Z_{\alpha\alpha}}, & Q \in \sigma_{\alpha\alpha} \\ 0, & \text{otherwise}, \end{cases} \tag{50}$$

where the normalization $Z_{\alpha\alpha}$ is given in (22). By construction, $\sum_{Q\in\sigma_{\alpha\alpha}} P_A(Q) = 1$. For general $n$, this becomes

$$\mathcal{Z}_n(Q) = \mathrm{Tr}_{\alpha\alpha}\left[\Pi_Q \rho_A^n\right] = \begin{cases} \frac{\chi_Q(q^n)}{Z_{\alpha\alpha}^n}, & Q \in \sigma_{\alpha\alpha} \\ 0, & \text{otherwise}. \end{cases} \tag{51}$$

The characters are $\chi_Q(q) = \frac{q^{h_Q}}{\eta(q)}$, where $h_Q$ are the conformal weights corresponding to the $U(1)$ charges in $\sigma_{\alpha\alpha}$, c.f. (35) and (37). Similar observations have been also made in [22], without providing explicit formulas and examples.

The fact that the form of the $\mathcal{Z}_n(Q)$ in all the considered $U(1)$ cases are the same can be understood as follows: They contain information about the partition function in a particular $U(1)$ sector and therefore are fully determined by the symmetry. The role of the boundary conditions is to determine which charges enter in the spectrum to begin with.

The entanglement entropy, given by (7), depends on the spectrum and therefore on the boundary conditions, c.f. section 3.2. In contrast, the form of the symmetry-resolved entanglement entropies only depend on the form of the characters $\chi_Q$ and therefore they have the same form independent of the boundary conditions, although the individual charges in the spectrum differ. Remember that here we focus on the same boundary conditions on both ends of the cylinder.

This new perspective also makes equipartition evident to *all* orders in the cutoff, independent of the boundary conditions. It follows directly from (51) for charges $Q \in \sigma_{\alpha\alpha}$ that

$$\begin{aligned} S_n^{\alpha\alpha}(Q) &= \frac{1}{1-n}\log\left[\frac{\mathcal{Z}_n(Q)}{(\mathcal{Z}_1(Q))^n}\right] = \frac{1}{1-n}\log\left[\frac{\eta^n(q)}{\eta(q^n)}\right] \\ &= \frac{W}{12}\frac{n+1}{n} + \frac{1}{1-n}\sum_{k=1}^{\infty}\log\left[\frac{(1-e^{-2Wk})^n}{1-e^{-2Wk/n}}\right] - \frac{1}{2}\log\left[\frac{W}{\pi}\right] + \frac{1}{2}\frac{\log n}{1-n}, \end{aligned} \tag{52}$$

In going to the second line, the conformal weights carrying the charge dependence have simply cancelled due to the explicit form of the characters, leading to equipartition of entanglement. Furthermore $\eta(\tilde{q}) = \sqrt{-i\tau}\,\eta(q)$ was used. The terms of order $W$ and $\log W$ in (52) have been first found in [20]. In this manuscript, we extend the analysis to all orders in the UV cutoff expansion, discussing the consequences of choosing different boundary conditions.

Note that the result in (52) does not explicitly depend on the compactification radius $R$, apart from the fact that $Q \in \sigma_{\alpha\alpha}$, where $\sigma_{\alpha\alpha}$ is dependent on $R$. Therefore, this result also holds for the non-compact free boson theory. A very important lesson from this section is that symmetry-resolved entanglement in the case of the free boson simply computes the entanglement of a single $U(1)$ character. It is hence no surprise

---

[3]There are no null vectors built purely from $a_n$ modes, which is a direct consequence of the algebra (26a). In contrast, representations built with Virasoro modes at $c = 1$ may indeed have null vectors [81,82].

that (52) coincides with the Rényi entropies of the non-compact free boson with DD boundaries, eq. (45), since that partition function consists only of a single character, which explains the occurence of the $\log W$ term.

Equipartiton of entanglement is also reflected in the Rényi entropies

$$
\begin{aligned}
S_n^{\alpha\alpha} &= \frac{1}{1-n} \log \left[ \frac{\eta^n(q)}{\eta(q^n)} \right] + \frac{1}{1-n} \log \left[ \frac{\sum_{Q \in \sigma_{\alpha\alpha}} q^{nh_Q}}{\left( \sum_{Q \in \sigma_{\alpha\alpha}} q^{h_Q} \right)^n} \right] \\
&= \frac{1}{1-n} \log \left[ \frac{\eta^n(q)}{\eta(q^n)} \right] + \frac{1}{1-n} \log \left[ \sum_{Q \in \sigma_{\alpha\alpha}} \left( P_A(Q) \right)^n \right] = S_{n,c}^{\alpha\alpha} + S_{n,f}^{\alpha\alpha}, \quad (53)
\end{aligned}
$$

where the expressions in terms of $q$ of (33) and (36) were used. The second summand stems from the $\hat{\mathfrak{u}}(1)$ primaries in the theory and, as explained in (7), it is the fluctuation Rényi entropy $S_{n,f}^{\alpha\alpha}$, accounting for fluctuations between charge sectors. Again, this result is exact to all orders in the cutoff and holds for *all* Rényi parameters $n$. The first term must thus be the configurational Rényi entropy $S_{n,c}^{\alpha\alpha}$, describing the entanglement in one charge sector. It accounts for the Fock space structure of the $\hat{\mathfrak{u}}(1)$ towers built on each primary state and is responsible for the well-known leading term of the Rényi entropy,

$$
\begin{aligned}
S_{n,c}^{\alpha\alpha} &= \frac{1}{1-n} \log \left[ \frac{\eta^n(q)}{\eta(q^n)} \right] = \frac{1}{1-n} \log \left[ \frac{\eta^n(\tilde{q})}{\eta(\tilde{q}^{\frac{1}{n}})} \frac{\sqrt{-in\tau}}{(\sqrt{-i\tau})^n} \right] \\
&= \frac{W}{12} \frac{n+1}{n} + \frac{1}{1-n} \sum_{k=1}^{\infty} \log \left[ \frac{(1 - e^{-2Wk})^n}{1 - e^{-2Wk/n}} \right] - \frac{1}{2} \log \left[ \frac{W}{\pi} \right] + \frac{1}{2} \frac{\log n}{1-n}, \quad (54)
\end{aligned}
$$

where $\eta(\tilde{q}) = \sqrt{-i\tau} \, \eta(q)$ has been used in the first and $\tau = i\pi/W$ and $\tilde{q} = e^{-2W}$ in the second line. Note that this expression is exact, and it is clear that performing $\epsilon \to 0$, i.e. $W \to \infty$, suppresses the infinite series, which stems from the $\hat{\mathfrak{u}}(1)$ descendants in the $\tilde{q}$ channel. Observe that $S_{n,c}^{\alpha\alpha} = S_n^{\alpha\alpha}(Q)$, i.e. the symmetry-resolved Rényi entropies account for the configurational Rényi entropies. As stressed in Sec. 2, the decomposition in the last step of (53) holds only in case of exact equipartition of entanglement entropies, which is what we find here for the compact massless free boson.

The $n \to 1$ limit of the configurational entropy $S_{n,c}^{DD}$ is straightforwardly taken in the expression (54). The full expression for the entanglement entropy is

$$
S_1 = \lim_{n \to 1} (S_{n,c}^{\alpha\alpha} + S_{n,f}^{\alpha\alpha}) = \lim_{n \to 1} \frac{1}{1-n} \log \left[ \frac{\eta^n(q)}{\eta(q^n)} \right] - \sum_{Q \in \sigma_{\alpha\alpha}} P_A(Q) \log \left[ P_A(Q) \right]. \quad (55)
$$

Note that the last two terms in (54) cancel out with terms of fluctuation entropy, once it is expressed in terms of $\tilde{q}$, at least for finite compactification radius. In the rest of this subsection we encounter an example where the fluctuation entropy is identically zero, and the final two terms indeed appear in the full entanglement entropy, perhaps contrary to expectations.

**Non-compact Free Boson: DD and NN boundary conditions:** It is illuminating to consider the decompactification limit $R \to \infty$. We begin with the DD case, for which the entire spectrum consists only of the $w = 0$ $\hat{\mathfrak{u}}(1)$ family, see (41). Thus $P_A(Q) = 1$ if $Q = Q_0^{(\Delta\varphi_0)}$ and zero otherwise, see (37). The Rényi entropies become

$$
S_n^{DD,(\infty)} = \frac{1}{1-n} \log \left[ \frac{Z_{D(\varphi_0'),D(\varphi_0)}^{(\infty)}(q^n)}{\left( Z_{D(\varphi_0'),D(\varphi_0)}^{(\infty)}(q) \right)^n} \right] = \frac{1}{1-n} \log \left[ \frac{\eta^n(q)}{\eta(q^n)} \right] = S_{n,c}^{DD}. \quad (56)
$$

There is no fluctuation entropy, $S_{n,f}^{DD} = 0$, as expected, since a second charge sector would be required for fluctuations to occur. Put another way, there is no uncertainty

in the charge measurement. No g-factor appears in the final result. It would, if one were to neglect the higher orders and approximate to keep only the leading terms in $\tilde{q}$, since it cancels out in the transformation from $\tilde{q}$ to $q$.

The result (56) has interesting consequences for the entanglement entropy,

$$S_1^{DD,(\infty)} = \frac{W}{6} - \frac{1}{2}\log\left[\frac{W}{\pi}\right] - \frac{1}{2} - \sum_{k=1}\left[\log\left(1 - e^{-2Wk}\right) - \frac{2Wk}{e^{2Wk} - 1}\right]. \qquad (57)$$

The second and third term is usually associated with the SREE for a charge sector $Q$. Here, however, it already appears in the regular entanglement entropy, since there are no contributions from the fluctuation entropy to cancel these terms.

The NN partition function of the non-compact boson includes, in contrast to the DD case, *all* $\hat{\mathfrak{u}}(1)$ families, c.f. (40). The configurational entropy remains as it is, and may be directly read off from (54). It remains to calculate the fluctuation Rényi entropy, which in this case is

$$S_{n,\mathrm{f}}^{NN,(\infty)} = \frac{1}{1-n}\log\left[\frac{\int_{\mathbb{R}} q^{n\alpha^2/(8\pi g)}d\alpha}{\left(\int_{\mathbb{R}} q^{\beta^2/(8\pi g)}d\beta\right)^n}\right] = \log \mathrm{g}_{N,(\infty)}^2 + \frac{1}{2}\log\left[\frac{W}{\pi}\right] - \frac{1}{2}\frac{\log n}{1-n}, \quad (58)$$

where $\mathrm{g}_{N,\infty} = (4\pi g)^{\frac{1}{4}}$. The second and third terms cancel in the Rényi entropy with terms of the configurational Rényi entropy (54) and in the limit $n \to 1$ the entanglement entropy reads,

$$S_1^{NN,(\infty)} = \frac{W}{6} + \log \mathrm{g}_{N,(\infty)}^2 - \sum_{k=1}\left[\log\left(1 - e^{-2Wk}\right) - \frac{2Wk}{e^{2Wk} - 1}\right]. \qquad (59)$$

Observe that, in contrast to the DD case, the g-factor now appears and the $\log W$ term is absent.

Before moving on to check these results using the charged moments, we comment on the case of mixed boundary conditions, i.e. DN and ND. While it is possible to calculate the entanglement entropy in these cases, it is not possible to resolve with respect to $U(1)$, because the mixed boundary conditions break this symmetry. This can be seen either from the boundary term required in the conservation equations of the $U(1)$ charges, as reviewed in appendix A, or from the modes $a_r$. Indeed, the modes acquire half-integer indices, $r \in \mathbb{Z} + \frac{1}{2}$, for mixed boundaries and therefore the conserved $U(1)$ charge $a_0$ is not defined. Therefore, resolution with respect to $U(1)$ is not possible in these cases, which can be rephrased by saying that (48) does not hold when $\alpha\beta = \mathrm{ND/DN}$, and $\mathcal{Q}$ given by the $U(1)$ generator. However, the ND and DN spectra still contain $\mathbb{Z}_2$ representations, with respect to which we resolve in section 4.2.

### 4.1.1 Cross-check: Compact Boson

In this section we confirm our previous results by employing the standard method in symmetry resolution, namely the $U(1)$ charged moments of the compact boson CFT. In order to calculate the charged moments to all orders, we employ the boundary state approach. We advocate for this approach as it allows to calculate to arbitrary orders in the UV cutoff. Therefore it is a vital addition to the usual symmetry resolution toolkit, which is based mainly on charged twist fields, which only allow to extract the leading terms. We note that this formalism has already been employed in [20], but has garnered little attention since.

The charged moments (13) for a $U(1)$ theory read

$$\mathcal{Z}_n^{\alpha\alpha}(\mu) = \frac{1}{Z_{\alpha\alpha}^n}\mathrm{tr}_{\mathcal{H}_{\alpha\alpha}}\left(\left(q^n\right)^{L_0 - \frac{c}{24}} e^{i\mu\mathcal{Q}}\right), \qquad (60)$$

where $\alpha$ still labels boundary conditions, $\mathcal{H}_{\alpha\alpha}$ is the corresponding Hilbert space. In the case of the free boson, (49) leads to the decomposition of the charged moments into *charged $U(1)$ characters*,

$$\mathcal{Z}_n^{\alpha\alpha}(\mu) = \frac{1}{Z_{\alpha\alpha}^n}\sum_{Q \in \sigma_{\alpha\alpha}} \chi_Q(q^n, \mu, 0), \qquad (61)$$

where $\sigma_{\alpha\alpha}$ is the spectrum of charges in the Hilbert space with boundary condition $\alpha$, given by (35) in the case of NN boundaries and (37) for DD boundaries. For a continuous spectrum, such as that of the non-compact boson, the sum becomes an integral. The *charged* characters are defined as

$$\chi_Q(q, \mu, u) = e^{i8\pi^2 gu} \text{tr}_{\mathcal{H}_Q^{\hat{u}(1)}} \left( q^{L_0 - \frac{c}{24}} e^{i\mu \mathcal{Q}} \right) = e^{i8\pi^2 gu} e^{i\mu Q} \frac{q^{h_Q}}{\eta(q)}. \tag{62}$$

The necessity of the phase parameter $u$ will become evident below.

This expression is unfortunately not useful in the $W \to \infty$ limit, in which $q \to 1^-$. In the $S$-dual frame on the other hand, the expansion variable $\tilde{q}$ tends to zero for $W \to \infty$. Charged moments can readily be computed in this frame by virtue of boundary states $\langle\!\langle \alpha \|$. For N boundaries these are (30a) and for D boundaries (30b). These allow for a straightforward expansion in orders of $\tilde{q}$. For $U(1)$ characters the $S$-modular transformation is represented by a Fourier transformation, and thus the charged characters transform as

$$\int \mathrm{d}Q' S_{Q,Q'} \chi_{Q'}(q, \mu, u) = \chi_Q \left( \tilde{q}, -\frac{\mu}{\tau}, u - \frac{1}{2\tau} \left( \frac{\mu}{2\pi} \right)^2 \right). \tag{63}$$

Here $S_{Q,Q'} = \frac{1}{\sqrt{4\pi g}} e^{i \frac{QQ'}{2g}}$ are the matrix elements of the $S$ transformation [83].

Looking at the definition (13) of the charged moments, we observe that they are essentially partition functions with an inserted exponential. Therefore, using the transformation law of the charged characters and (22), the charged moments can be expressed in terms of the boundary states, in this case given by (30). The charged moments read

$$\mathcal{Z}_n^{\alpha\alpha}(\mu) = \frac{1}{Z_{\alpha\alpha}^n} \tilde{q}^{\frac{1}{n} 2\pi g \left( \frac{\mu}{2\pi} \right)^2} \langle\!\langle \alpha \| \left( \tilde{q}^{\frac{1}{n}} \right)^{L_0 - \frac{c}{24}} e^{-i \frac{\mu}{n\tau} \mathcal{Q}} \| \alpha \rangle\!\rangle. \tag{64}$$

Calculating the charged moments using (64) yields

$$\mathcal{Z}_n^{DD}(\mu) = \frac{1}{Z_{DD}^n} \sum_{Q \in \sigma_{DD}} \frac{1}{\eta(q^n)} q^{\frac{n}{8\pi g} Q^2} e^{i\mu Q}, \tag{65a}$$

$$\mathcal{Z}_n^{NN}(\mu) = \frac{1}{Z_{NN}^n} \sum_{Q \in \sigma_{NN}} \frac{1}{\eta(q^n)} q^{\frac{n}{8\pi g} Q^2} e^{i\mu Q}. \tag{65b}$$

We note that similar expressions were already presented in [20], albeit restricted to only two values of $\Delta\varphi_0$ and $\Delta\theta_0$. Here, we generalize to all possible values of $\Delta\varphi_0$ and $\Delta\theta_0$.

To calculate the $\mathcal{Z}_n(Q)$, the charged moments have to be Fourier transformed. In the compact case a subtlety arises, since the values of $Q$ are neither integer nor continuous for finite, non-zero compactification radius. By standard Fourier theory this leads to all functions of $\mu$ being periodic with period $2\pi R$ and $\frac{4\pi}{R}$, respectively. By rescaling the integration variable and substitution in the Fourier transformation, the new function can be made to be $2\pi$ periodic. For both boundary conditions,

$$\mathcal{Z}_n^{\alpha\alpha}(Q) = \frac{1}{Z_{\alpha\alpha}^n} \frac{q^{n \frac{Q^2}{8\pi g}}}{\eta(q^n)}, \qquad Q \in \sigma_{\alpha\alpha}. \tag{66}$$

This result agrees with our previous result in (51).

The difference between $DD$ and $NN$ boundary conditions again is that the charges in the spectrum take different values depending on the compactification radius $R$.

### 4.1.2 Cross-check: Non-compact Boson

A cross-check similar to the one discussed in the previous subsection can be done for the non-compact boson theory. In this case the boundary states (30) are replaced by

$$\| N \rangle\!\rangle_\infty = \mathsf{g}_{N,\infty} |0\rangle\!\rangle, \tag{67a}$$

$$\|D(\varphi_0)\rangle\!\rangle_\infty = \mathtt{g}_{D,\infty} \int_{\mathbb{R}} \mathrm{d}Q e^{-iQ\varphi_0} |Q\rangle\!\rangle. \tag{67b}$$

Here $|Q\rangle\!\rangle$ are the Ishibashi states, c.f. (31), and $\varphi_0$ remains the boundary value of the field $\varphi$. Note that only a single Ishibashi state contributes in the NN case, c.f. section 3.1. The g-factors are now

$$\mathtt{g}_{D,\infty} = \langle 0\|D(\varphi_0)\rangle\!\rangle_\infty = (4\pi g)^{-\frac{1}{4}}, \tag{68}$$

$$\mathtt{g}_{N,\infty} = \langle 0\|N\rangle\!\rangle_\infty = (4\pi g)^{\frac{1}{4}}. \tag{69}$$

In the case of NN boundary conditions,

$$\mathscr{Z}_n^{NN}(\mu) = \mathtt{g}_{N,\infty}^2 \frac{\tilde{q}^{\frac{1}{n}2\pi g\left(\frac{\mu}{2\pi}\right)^2}}{Z_{NN}^n \eta(\tilde{q}^{\frac{1}{n}})} = \frac{\mathtt{g}_{N,\infty}^2}{Z_{NN}^n} \chi_0\left(\tilde{q}^{\frac{1}{n}}, -\frac{\mu}{n\tau}, -\frac{1}{2}\left(\frac{\mu}{2\pi}\right)^2\right). \tag{70}$$

This result is exact to all orders of $\epsilon$. In the $S$-dual picture, i.e. in the $\tilde{q}$ expression, only one of the bulk modes is allowed to propagate for NN boundary conditions. Since the $S$ transformation acts as a Fourier transform, we expect that all BCFT modes are allowed to propagate, which can be seen from the fact that the matrix elements $S_{0,Q'} = \frac{1}{\sqrt{4\pi g}}$ are independent of $Q'$. Therefore all $U(1)$ representations contribute equally in the partition function and the charged moments and the expression in terms of $q$ reads

$$\mathscr{Z}_n^{NN}(\mu) = \frac{1}{Z_{NN}^n} \int \mathrm{d}Q \chi_Q\left(q^n, \mu, 0\right), \tag{71}$$

where $\chi_Q\left(q^n, \mu, 0\right)$ is defined in (62). Furthermore we see from (70) that the leading order in $\tilde{q}$ produces the charged moments computed in the twist field picture and the holographic calculation [63],

$$\mathscr{Z}_n^{NN}(\mu) \sim \frac{\mathtt{g}_{N,\infty}^2}{Z_{NN}^n} \left(\tilde{q}^{\frac{1}{n}}\right)^{2\pi g\left(\frac{\mu}{2\pi}\right)^2 - \frac{c}{24}}. \tag{72}$$

Fourier transforming the exact result (70), we obtain for $Q \in \mathbb{R}$

$$\mathscr{Z}_n^{NN}(Q) = \frac{1}{Z_{NN}^n} \frac{q^{n\frac{Q^2}{8\pi g}}}{\eta(q^n)} = \frac{1}{Z_{NN}^n} \chi_Q(q^n, 0, 0). \tag{73}$$

This result is consistent with (51). Observe that the boundary states automatically contain the information about which charges enter the spectrum, c.f. the discussion in section 3.1. In the case of DD boundary conditions the analysis of the boundary states yields the charged moments

$$\mathscr{Z}_n^{DD}(\mu) = \frac{\mathtt{g}_{D,\infty}^2}{Z_{DD}^n} \tilde{q}^{\frac{1}{n}2\pi g\left(\frac{\mu}{2\pi}\right)^2} \int \mathrm{d}Q e^{-iQ\Delta\varphi_0} e^{-i\frac{\mu}{n\tau}Q} \frac{\tilde{q}^{\frac{1}{n}\frac{Q^2}{8\pi g}}}{\eta(\tilde{q}^{\frac{1}{n}})}. \tag{74}$$

In contrast to the NN case all bulk modes propagate in the $\tilde{q}$ expression. In the BCFT picture, the charged moments read

$$\mathscr{Z}_n^{DD}(\mu) = \frac{1}{Z_{DD}^n} \frac{1}{\eta(q^n)} e^{i\mu 2g\Delta\varphi_0} q^{n2\pi g\left(\frac{\Delta\varphi_0}{2\pi}\right)^2} = \frac{1}{Z_{DD}^n} \chi_{2g\Delta\varphi_0}(q^n, \mu, 0). \tag{75}$$

Instead of consisting of a single bulk mode in $\tilde{q}$ expression, the charged moments consist of a single mode in the $q$ expression, again c.f. section 3.1.

Fourier transforming the charged moments gives

$$\mathscr{Z}_n^{DD}(Q) = \frac{1}{Z_{DD}^n} \frac{q^{n\frac{Q^2}{8\pi g}}}{\eta(q^n)} \delta\left(Q - 2g\Delta\varphi_0\right). \tag{76}$$

This is again consistent with (51). The Dirac distribution appearing in this result corroborates that the spectrum of the free boson with DD boundary conditions consists of only one $\hat{\mathfrak{u}}(1)$ representation.

We find it worth stressing again that (76), (73) and (66) confirm the discussion reported in section 4.1, where we pointed out that the charged partition functions $\mathcal{Z}(Q)$ have the same expression in all the $U(1)$ and $\mathbb{R}$ cases. Different boundary conditions and considering the compact or the non-compact abelian symmetry group only determine which charges enter in the spectrum.

## 4.2   $\mathbb{Z}_2$ Symmetry Resolution

We now turn our attention to resolution with respect to the simplest finite discrete group, namely $\mathbb{Z}_2$. The DD, NN and ND partition functions carry $\mathbb{Z}_2$ representations so that their entanglement structure can be unveiled. The reader will not be surprised to find that there is a close relationship with the conventional $\mathbb{Z}_2$ orbifold of the bosonic theory [81]. For all the boundary conditions considered here we compute the charged partition function (10) directly from the knowledge of the projectors in the two charge sectors. This is in the spirit of section 4.1. As a consequence, we do not compute and exploit the charged moments as in (14) and (15) with $N = 2$.

### 4.2.1   $\mathbb{Z}_2$ Symmetry Resolution for DD and NN

While treating the DD and NN cases, we restrict to the simple cases $\Delta\varphi_0 = 0 = \Delta\theta_0$. It is then also useful to introduce a common notation for the primaries in both boundary theories, we call them $|s\rangle$, $s \in \mathbb{Z}$ with $a_0 |s\rangle = Q_s |s\rangle$, where $Q_s$ stands for either $Q_w^{(0)}$ from (37) or $Q_m^{(0)}$ from (35).

Denoting $\mathbb{Z}_2 = \{e, g\}$, where $e$ is the unit element, the non-trivial element $g$ acts on the Fock modes via $g^{-1}a_k g = -a_k$. For $k = 0$ this implies the $\mathbb{Z}_2$ action flips the sign of the $U(1)$ charge, $Q_s \to -Q_s$, and so we can identify $g |s\rangle = |-s\rangle$.

A projector onto $g = \pm 1$ eigenspaces, required for (10), is $\Pi_\pm = \frac{e \pm g}{2}$. We are thus led to evaluate traces of the type $\text{Tr}_{\mathcal{H}_{\alpha\alpha}}[gq^{L_0}]$, where $\mathcal{H}_{\alpha\alpha}$ is the Hilbert space of either the DD or NN theory. For their evaluation it is convenient to reorganize the spectrum in states of the form

$$a_{k_1} a_{k_2} \ldots a_{k_l} \left( |s\rangle \pm |-s\rangle \right), \tag{77}$$

which are eigenstates of $g$. For a fixed configuration of modes $a_j$, these pairs of states contribute with opposite sign to $\text{Tr}_{\mathcal{H}_{\alpha\alpha}}[gq^{L_0}]$ and thus cancel away, as long as $s \neq 0$. This rearrangement of the Hilbert space has the following physical meaning. Each of the $U(1)$ sectors of the theory discussed in section 4.1 contains a $\mathbb{Z}_2$-even and a $\mathbb{Z}_2$-odd part. Thus, in order to access the two $\mathbb{Z}_2$ sectors it is necessary to decompose the $U(1)$ sectors and reorganize them, as done by the change of basis (77).

What remains is to evaluate the trace in the $s = 0$ sector, which is done by elementary methods [81],

$$\text{Tr}_{\mathcal{H}_{\alpha\alpha}}\left[ gq^{L_0 - \frac{1}{24}} \right] = \text{Tr}_{s=0}\left[ gq^{L_0 - \frac{1}{24}} \right] = q^{-\frac{1}{24}} \prod_{k=1}^{\infty} \frac{1}{1 + q^k} = \sqrt{2\frac{\eta(q)}{\theta_2(q)}} = \sqrt{2\frac{\eta(\tilde{q})}{\theta_4(\tilde{q})}}, \tag{78}$$

where $\theta_i$ are modular Jacobi theta functions. A summary of their properties can be found in appendix B.

The trace (10) for the two possible $\mathbb{Z}_2$ representations, labelled $\pm$, can thus be evaluated

$$\mathcal{Z}_n^{\alpha\alpha}(\pm) = \text{Tr}_{\mathcal{H}_{\alpha\alpha}}[\Pi_\pm \rho_A^n]$$

$$= \frac{1}{(Z_{\alpha\alpha}(q))^n} \text{Tr}_{\mathcal{H}_{\alpha\alpha}}\left[ \frac{e \pm g}{2} q^{n(L_0 - \frac{1}{24})} \right]$$

$$= \frac{1}{2(Z_{\alpha\alpha}(q))^n} \left( Z_{\alpha\alpha}(q^n) \pm \sqrt{2\frac{\eta(\tilde{q}^{1/n})}{\theta_4(\tilde{q}^{1/n})}} \right)$$

$$
= \frac{1}{2(Z_{\alpha\alpha}(q))^n} \left( \mathbf{g}_\alpha^2 \frac{e^{2W/(24n)}}{\prod_{k=1}^\infty (1 - e^{-2Wk/n})} \sum_{s \in \mathbb{Z}} e^{-2Wh_s/n} \right.
$$
$$
\left. \pm \sqrt{2} \frac{e^{-W/(24n)}}{\prod_{k=1}^\infty (1 - e^{-W(2k-1)/n})} \right), \tag{79}
$$

with $h_s = Q_s^2/(8\pi g)$. When $W \to \infty$, the first summand goes with $e^{\frac{W}{12n}}$, while the second term decays rapidly with $e^{-\frac{W}{24n}}$. Without computing the symmetry-resolved entanglement form these replica partition functions, we can thus already infer that equipartition holds at leading order, thereby confirming the results of [32]. When including all orders, however, it is clear that equipartition cannot hold, since (79) depends on the chosen sign. Observe that $\mathcal{Z}_1(+)$ is proportional to the holomorphic part of the projected untwisted sector of the conventional $\mathbb{Z}_2$ orbifold [81].

Finally, the symmetry-resolved Rényi entropies are

$$
S_n^{\alpha\alpha}(\pm) = \frac{1}{1-n} \log\left[ \frac{\mathcal{Z}_n^{\alpha\alpha}(\pm)}{(\mathcal{Z}_1^{\alpha\alpha}(\pm))^n} \right]. \tag{80}
$$

When either $\Delta\varphi_0 = 0$ or $\Delta\theta_0 = 0$, we can expand (80) as

$$
S_n^{\alpha\alpha}(\pm) = \frac{1+n}{12n} W - \ln 2 + \ln \mathbf{g}_\alpha^2 + \dots, \tag{81}
$$

where the dots denote the subleading corrections which vanish exponentially in $W$ as the UV cutoff goes to zero, and are responsible for the breaking of the entanglement equipartition. The term $\ln 2$ is nothing but the logarithm of the number of $\mathbb{Z}_2$ sectors, and is the leading contribution to the fluctuation entropy. In other words, no double logarithmic corrections arise in the $\mathbb{Z}_2$ symmetry-resolution of the entanglement [32].

### 4.2.2   $\mathbb{Z}_2$ Symmetry Resolution for DN

It is convenient rewrite the ND BCFT partition function (38) as

$$
Z_{ND}(q) = \sqrt{\frac{\eta(q)}{\theta_4(q)}}, \tag{82}
$$

which makes evident that this spectrum is equivalent to the holomorphic part of the unprojected twisted sector of the bosonic $\mathbb{Z}_2$ orbifold found in [81]. It carries $\mathbb{Z}_2$ representations with respect to which we wish to resolve. To that end, it is convenient to split (39) into $g = \pm 1$ eigenspaces,

$$
\mathcal{H}_{DN}^{(+)} = \{ a_{-r_1} \dots a_{-r_{2k}} |\sigma\rangle \}, \tag{83}
$$
$$
\mathcal{H}_{DN}^{(-)} = \{ a_{-r_1} \dots a_{-r_{2k+1}} |\sigma\rangle \}, \tag{84}
$$

with $r_i \in \mathbb{Z} + \frac{1}{2}$.
The projector onto these eigenspaces is still $\Pi_\pm = \frac{e+g}{2}$, and we consider

$$
\mathrm{Tr}_{\mathcal{H}_{DN}} [g q^{L_0 - \frac{1}{24}}] = \sqrt{\frac{\eta(\tilde{q})}{\theta_3(\tilde{q})}}, \tag{85}
$$

whose derivation can be found in [81]. The replica partition function (10) is thus

$$
\mathcal{Z}_n^{DN}(\pm) = \frac{1}{2(Z_{DN}(q))^n} \left( \sqrt{\frac{\eta(q^n)}{\theta_4(q^n)}} \pm \sqrt{\frac{\eta(q^n)}{\theta_3(q^n)}} \right)
$$
$$
= \frac{1}{2(Z_{DN}(q))^n} \left( \sqrt{\frac{\eta(\tilde{q}^{1/n})}{\theta_2(q^{1/n})}} \pm \sqrt{\frac{\eta(\tilde{q}^{1/n})}{\theta_3(\tilde{q}^{1/n})}} \right)
$$

$$= \frac{1}{2(Z_{DN}(q))^n} \left( \mathsf{g}_N \mathsf{g}_D \, e^{W/(12n)} \prod_{k=1}^{\infty} (1 - e^{-2Wk/n})^{-1} \right.$$
$$\left. \pm \prod_{k=1}^{\infty} (1 + e^{-W(2k-1)/n})^{-1} \right). \tag{86}$$

where we used $\mathsf{g}_D \mathsf{g}_N = 1/\sqrt{2}$, see (32). Again it is clear that for $W \to \infty$, the first term dominates and hence equipartition is guaranteed at leading order. Similar to before, equipartition breaks down once all orders are included. Observe that $\mathcal{Z}_1(+)$ is proportional to the holomorphic part of the projected twisted sector of the conventional $\mathbb{Z}_2$ orbifold [81].

Finally, the symmetry-resolved Rényi entropies are

$$S_n^{ND}(\pm) = \frac{1}{1-n} \log \left[ \frac{\mathcal{Z}_n^{DN}(\pm)}{\left( \mathcal{Z}_1^{DN}(\pm) \right)^n} \right]. \tag{87}$$

For the leading orders one finds

$$S_n^{ND}(\pm) = \frac{1+n}{12n} W - \ln 2 + \ln(\mathsf{g}_D \mathsf{g}_N) + \dots, \tag{88}$$

where the dots have the same meaning, and the $\ln 2$ the same interpretation discussed below (81).

# 5 Conclusions

## 5.1 Discussion

In this article we investigated the symmetry resolution of entanglement in 2D CFTs by employing the BCFT approach for the computation of the entanglement entropies [12, 13]. More concretely, any factorization of Hilbert space associated with spatial domains requires the imposition of boundary conditions at the entangling surface in a QFT, see (16). These boundary conditions determine the field content in the entanglement spectrum. If, furthermore, the systems is governed by a symmetry group $G$, said entanglement spectrum decomposes into charge sectors labelled by irreducible representations of $G$. In this paper we showed that symmetry resolution with respect to $G = U(1), \mathbb{R}, \mathbb{Z}_2$ is achieved in the free boson CFT by extracting the characters of said irreducible representations, c.f. (10) and (51), and by computing their entanglement entropy (11). We found that this approach has a number of advantages:

- It bypasses the potentially laborious computation of charged moments (13).

- In contrast to the charged twist field approach [19], which only provides the leading contributions in a UV cutoff expansion, the setup here provides symmetry-resolved entanglement entropies to all orders in the UV cutoff. This is similar to the situation with regular twist fields, which provide the $cW/6$ term in the entanglement entropy, but not the higher lying terms described in [12].

- It provides conceptual insight. For instance, it explains the origin of equipartition of entanglement, if present. This is a simple consequence of the structure of the characters of the symmetry group. Equipartition has been studied in the case of $U(1)$ before in [20] and for $\mathbb{Z}_2$ in [32]. In both cases equipartition had been established to leading orders in the UV cutoff. Our setup allows us to extend their analysis to all orders and confirm $U(1)$ equipartition (52), while we find breaking of $\mathbb{Z}_2$ equipartition, c.f. (79) and (86).

Another aspect we find worth stressing is the role of boundary conditions imposed on the annulus geometry. Typically, the boundary conditions considered in BCFT are conformal boundary conditions, which deny energy-momentum flow across the

boundary. From the point of view of symmetries, this means that the holomorphic and the anti-holomorphic Virasoro algebrae reduce to a single Virasoro algebra. If the theory has an additional global symmetry, the boundary conditions may break this symmetry. If we want to compute the symmetry-resolved entanglement with respect to a given group $G$, we must choose boundary conditions that preserve such a group. In the cases considered in this article, we observed that NN and DD boundary conditions preserve one of the two $U(1)$ (or $\mathbb{R}$) symmetries of the theory without boundaries. The $U(1)$ preserved in the NN case is different from the one preserved in the DD case, c.f. appendix A for more details. In both cases, a symmetry resolution of the entanglement entropies can be achieved in the BCFT setup. On the other hand, when we impose ND or DN boundary conditions, the only residual symmetry is $\mathbb{Z}_2$ and therefore it is not possible to resolve the spectrum of the BCFT, i.e. the entanglement spectrum, with respect to $U(1)$ (or $\mathbb{R}$). In this latter case, the partition function of the BCFT is expanded in terms of the characters of twisted $U(1)$ representations [78, 80]. Thus, even if there is no $U(1)$ charge present for ND or DN boundary conditions, the theory bears remnants of the $U(1) \times U(1)$ symmetry present without boundaries, i.e. before the mapping the theory to the annulus.

## 5.2   Summary of Results

We have focused on the cases where the target space of the bosonic field is a circle (symmetry group $U(1)$) or non-compact (symmetry group $\mathbb{R}$). In both cases, we computed the charged partition function $\mathcal{Z}_n(Q)$ from the characters contained in the BCFT partition function, c.f. (51) for generic $n$ and in (50) for $n = 1$. The corresponding symmetry-resolved Rényi entropies are given in (52). We applied the same procedure for the resolution with respect to the $\mathbb{Z}_2$ symmetry. For this case, the charged partition functions are given in (79) for NN or DD boundary conditions, and in (86) for ND boundaries. The corresponding symmetry-resolved Rényi entropies are straightforwardly computed using these results. We now provide a detailed discussion of each of these results.

Our first example is the compact boson resolved with respect to the global $U(1)$ symmetry. The symmetry-resolved entropies for this theory have been first considered in [20], where the leading order contributions in the UV cutoff expansion were computed and found to be independent of the choice of boundary conditions on the annulus. We extend the analysis of [20] by calculating all the possible higher-order terms for all allowed combinations of boundary conditions. We in particular considered two choices of boundary conditions, DD and NN, both of which allowing for a residual $U(1)$ symmetry. Our results (52) exhibit some very interesting features: First, the equipartition of entanglement holds to all orders in the cutoff expansion. This is an extension of the results in [20], which reported equipartition to leading order. Our result originates from the explicit form of the $U(1)$ characters $\chi_Q(q) = q^{h_Q}/\eta(q)$. This particular form implies that $h_Q$ cancels out in the ratio $\mathcal{Z}_n(Q)/(\mathcal{Z}_1(Q))^n$, so that no charge dependence remains, as is seen in (50) and (51). This is the first main result of the paper. Moreover, the explicit expressions of the symmetry-resolved entanglement entropies are the same for both the NN and DD case. The only difference lies in the allowed values of the charges $Q$, which are determined by the choice of the boundary conditions.

As a further example, we considered the decompactification regime of the compact free boson theory, namely the limit $R \to \infty$. The resulting theory has a symmetry given by the group $\mathbb{R}$. Thus, its symmetry resolution can be regarded as a first simple example of symmetry-resolved entanglement in presence of a non-compact symmetry group. Also in this case the expression for the symmetry-resolved entropies (52) only depends on the charge allowed by the boundary conditions. We find it worthwhile to stress an interesting feature of the non-compact boson with DD boundary conditions: its spectrum consists only of a single charge sector and therefore the only non-vanishing symmetry-resolved entanglement entropies coincide with the total entanglement entropies. This is somewhat surprising, since it shows that the double

logarithmic correction in the symmetry-resolved entanglement entropies is also present in the full entanglement entropies, which, to our knowledge, has not been reported so far.

The BCFT approach also allowed us to compute the entanglement entropies resolved with respect to a $\mathbb{Z}_2$ symmetry. These results are reported in (80) for NN and DD boundary conditions. Interestingly, in presence of DN and ND boundary conditions the $\mathbb{Z}_2$ symmetry is still present and therefore the symmetry-resolved entropies can be computed. They are given in (87). Again, these results are exact in the cutoff expansion, and reduce to the findings of [32] to leading order in the cutoff expansion. Our results for $U(1)$ and $\mathbb{R}$ have been cross-checked in sections 4.1.1 and 4.1.2 by computing first the charged moments and then performing the Fourier transform.

## 5.3 Outlook

Based on the findings of our work, there are many future directions to pursue: First, we have come across twisted $U(1)$ representations in the case of mixed boundary conditions. These arise from a twisted $U(1)$ and retain some of the features of the untwisted case. For instance, the structure of the $\hat{u}(1)$ Kac-Moody algebra persists, in that the modes still satisfy $[a_r, a_s] = -r\delta_{r+s,0}$ where the indices are half-integral. In the picture put forward in section 4, symmetry-resolution is achieved by isolating a character. We can therefore resolve with respect to these characters, i.e. with respect to the twisted $U(1)$ symmetry. Nevertheless it is not clear what the physical charge operator is for this representation. In particular, as shown in appendix A, Noether's procedure does not provide a conserved charge operator in this case. The identification of such charges is therefore an interesting route to take. This question is by no means of pure mathematical nature, as physical systems have already been identified which feature ND boundaries at their entangling points, see [84]. We point out that for ND/DN boundaries, the entanglement spectrum consistst of only a single twisted character. In this regard, the ND case is very similar to the case of DD boundaries for the non-compact boson.

Second, the free boson is in many regards a very simple theory, and therefore ideal as a toy model. In order to test the utility of our approach, it is thus necessary to find other, more involved theories with global symmetries, potentially non-abelian, in which our approach of section 4 can be applied. One obvious candidate are Wess-Zumino-Witten models with global symmetry $G$. Results already exist for these models [22], in which the entanglement spectrum was resolved up to $\mathcal{O}(\epsilon)$ in the UV cutoff $\epsilon$ with respect to representations of $G$. Our approach makes evident how to resolve with respect even to subgroups of $H \subset G$. What is required is to decompose representations of $G$ into those of $H$. Going one step further using character decompositions, it will even be possible to resolve coset models, for which no results exist at present. Such coset models appear in higher spin holography [65, 85, 86], as well as in exactly solvable top-down models of $\text{AdS}_3/\text{CFT}_2$ [87]. Therefore our approach applies to a large number of CFTs, and does so to all orders in the UV cutoff.

Third, One of the abelian symmetry groups considered in this work is the group $\mathbb{R}$, which is non-compact. The results discussed in Sec. 4 provide a first entanglement resolution with respect to a non-compact, albeit simple, symmetry. Along this line, it would be interesting to consider the resolution of entanglement in presence of non-abelian, non-compact groups. A promising candidate for this purpose is $SL(2, \mathbb{R})$, whose irreducible representations, crucial for applying the BCFT approach, are known [88].

Fourth, the free fermion CFT is a natural candidate for the application of our approach. The additional lessons to be learned here come from the spin structure, i.e. anti-periodic and periodic boundary conditions which are to be imposed on top of Neumann and Dirichlet conditions. In [36] it has been observed that the symmetry-resolved entanglement entropies depend on the chosen spin sector. Moreover, this dependence induces subleading corrections breaking the equipartition. It would be then interesting to understand more about the interplay between the spin sectors

and the boundary conditions to be imposed on the annulus geometry in the BCFT approach. Moving further this may allow to resolve supersymmetric models.

Fifth, entanglement is of major importance, via the AdS/CFT correspondence, for the understanding of space-time itself [89, 90]. First results on the symmetry resolution of entanglement in holography already exist [62–66, 91]. All these works are based on the bottom-up approach to holography. In contrast, given a top-down model for which the D brane states are known, it will be possible by using our method to resolve these theories with respect to their symmetry algebra, to all orders in the UV cutoff, and also in the bulk Newton constant. Importantly, such studies might provide a geometric interpretation of the non-leading order terms in the UV cutoff and the bulk Newton constant. The leading order of the entanglement entropy "builds spacetime" [90], and the higher orders add quantum effects to the classical spacetime geometry [92, 93]. A possible starting point would be to compare our results for the free boson with the bulk entanglement calculation of [94].

Finally, analyses relating the entanglement properties of quantum chains in the continuum limit to boundary conditions of BCFT models have been performed for the harmonic chain [84], for the Ising chain, and also out-of-equilibrium [11, 95]. It is interesting to apply our method in these cases since they allow direct comparison with simulations. Such an investigation can potentially allow to separate all lattice contributions from those of the CFT, providing an estimate for how many orders in UV cutoff of the CFT result should be trusted in practical applications.

# Acknowledgements

We thank Filiberto Ares, Johanna Erdmenger, Marius Gerbershagen, Sara Murciano, Ronny Thomale, and Konstantin Weisenberger for useful discussions. The work of all authors was funded by the Deutsche Forschungsgemeinschaft (DFG, German Research Foundation) through Project-ID 258499086—SFB 1170 "ToCoTronics" and through the Würzburg-Dresden Cluster of Excellence on Complexity and Topology in Quantum Matter – ct.qmat Project-ID 390858490—EXC 2147. S.Z. acknowledges financial support by the China Scholarship Council. Christian Northe furthermore acknowledges support from the Israel Science Foundation (grant No. 1417/21).

# A $U(1)$ symmetry breaking by ND and DN boundaries

In this appendix we explain how boundaries affect the $U(1) \times U(1)$ symmetry of the bosonic CFT. We show how NN and DD boundaries select a particular embedding of a conserved $U(1)$ generator into the original symmetry and move on to show how mixed boundaries deny the presence of such an operator.

A free massless bosonic field on a two-dimensional Euclidean manifold possesses two conserved currents and their respective charges. The first is the Noether current and the second a topological current

$$j^\mu = \partial^\mu \varphi, \qquad j^\mu_{\text{top}} = \epsilon^{\mu\nu} j_\nu. \tag{89}$$

The Noether current is divergence free, $\partial_\mu j^\mu = 0$ by virtue of the equations of motion while the topological current is divergence free by construction, $\partial_\mu j^\mu_{\text{top}} = 0$. Furthermore, they induce two natural charge operators

$$\mathcal{Q} = \int j^\tau d\sigma, \qquad \mathcal{Q}_{\text{top}} = \int j^\tau_{\text{top}} d\sigma. \tag{90}$$

where the spatial coordinate $\sigma$ is integrated over its full domain. Equations (89) are local conservation relations. Global properties are investigated by integrating their

respective divergences on a spatial domain. For the Noether current this results in

$$0 = \int \partial_\mu j^\mu d\sigma = \partial_\tau \mathcal{Q} + j_\sigma\big|_{bdy} = \partial_\tau \mathcal{Q} + \partial_\sigma \varphi\big|_{bdy}. \tag{91}$$

On a manifold without boundary, the second term on the right hand side can be neglected and as a result $\mathcal{Q}$ is conserved. On a manifold with boundary, however, $\mathcal{Q}$ is only conserved in presence of Neumann boundary conditions (28). Taking the solution of the equations of motion for the bosonic angular variable on a strip geometry[4] with $\tau \in \mathbb{R}$ and $\sigma \in [0, \pi]$ and Neumann boundaries at $\sigma = 0, \pi$,

$$\varphi_{NN}(\tau, \sigma) = \varphi_0 - i\frac{\tau}{\sqrt{\pi g}}a_0 + \frac{i}{\sqrt{\pi g}}\sum_{k \neq 0}\frac{1}{k}a_k e^{-k\tau}\cos(k\sigma), \qquad \partial_\sigma \varphi_{NN}\big|_{\sigma=0,\pi} = 0, \tag{92}$$

it may easily be verified from (90) that $\mathcal{Q} \propto a_0$.

Similarly, noting that $\partial_\mu j^\mu_{\text{top}} = 0$ implies $\partial_\tau j_\sigma = \partial_\sigma j_\tau$, the temporal derivate of the topological charge can be evaluated,

$$\partial_\tau \mathcal{Q}_{\text{top}} = \int \partial_\tau j^\tau_{\text{top}} d\sigma = \int \partial_\tau j^\sigma d\sigma = j^\tau\big|_{bdy} = \partial_\tau \varphi\big|_{bdy}. \tag{93}$$

On a manifold without boundary, the right hand side can be neglected and as a result $\mathcal{Q}_{\text{top}}$ is conserved. On a manifold with boundary, however, $\mathcal{Q}_{\text{top}}$ is only conserved in presence of Dirichlet boundary conditions (29). Taking the solution of the equations of motion for the bosonic angular variable on a strip geometry with Dirichlet boundaries at $\sigma = 0, \pi$,

$$\varphi_{DD}(\tau, \sigma) = \varphi_0 + \frac{\sigma}{\sqrt{\pi g}}a_0 + \frac{1}{\sqrt{\pi g}}\sum_{k \neq 0}\frac{1}{k}a_k e^{-k\tau}\sin(k\sigma), \tag{94}$$

it may easily be verified from (90) that $\mathcal{Q}_{\text{top}} \propto a_0$.

On a manifold without boundary the charges $\mathcal{Q}$ and $\mathcal{Q}_{\text{top}}$ generate the $U(1) \times U(1)$ symmetry of the free boson theory. When imposing a boundary, the symmetry is broken down. In principle, all symmetry can be broken by a boundary. Special classes of $U(1)$ symmetry-preserving boundaries can be chosen, such as Neumann and Dirichlet boundary conditions. These boundaries embed a new $U(1)$ symmetry into $U(1) \times U(1)$. The generator of the embedded $U(1)$ is a superposition of the $U(1) \times U(1)$ charges,

$$\mathcal{Q}_{\text{bdy}} = a\mathcal{Q} + b\mathcal{Q}_{\text{top}}, \tag{95}$$

where $a, b \in \mathbb{R}$ and never vanish simultaneously. Taking a temporal derivative,

$$\partial_\tau \mathcal{Q}_{\text{bdy}} = -a\,\partial_\sigma \varphi\big|_{bdy} + b\,\partial_\tau \varphi\big|_{bdy} = -a\,\partial_\sigma \varphi\big|_{\sigma=0}^{\sigma=\pi} + b\,\partial_\tau \varphi\big|_{\sigma=0}^{\sigma=\pi}, \tag{96}$$

reveals how distinct boundary conditions embed a $U(1)$ symmetry into $U(1) \times U(1)$. This used (91) and (93) and in going to the last expression we have chosen the manifold to be a strip. Clearly, for $(a = 1, b = 0)$ conservation of $\mathcal{Q}_{\text{bdy}}$ requires Neumann boundaries, while $(a = 0, b = 1)$ selects Dirichlet conditions.

Turning to mixed boundaries, we choose to have a Neumann condition at $\sigma = \pi$ and a Dirichlet condition at $\sigma = 0$. This yields

$$\partial_\tau \mathcal{Q}_{\text{bdy}} = a\,\partial_\sigma \varphi(\sigma = 0) + b\,\partial_\tau \varphi(\sigma = \pi). \tag{97}$$

No parameters $a, b$ can be chosen to have this expression vanish. Hence no conserved $U(1)$ charge can be embedded into the original $U(1) \times U(1)$ symmetry. This may be checked explicitly using the solution for the free boson,

$$\varphi_{DN}(\tau, \sigma) = \varphi_0 - \frac{1}{\sqrt{\pi g}}\sum_{r \in \mathbb{Z} + \frac{1}{2}}\frac{1}{r}a_r e^{-r\tau}\sin(r\sigma). \tag{98}$$

---

[4]Note that the arguments presented in this appendix are independent of the choice of manifold and demonstrate how boundaries affect the $U(1)$ symmetry of a theory. In particular, our discussion also applies to the annulus, which is used throughout the main text. Here we choose the strip since it allows for simple illustration of the charges via the bosonic solutions (92), (94), (98) and (99).

The same conclusion is obtained for the reversed case of a Neumann condition at $\sigma = 0$ and a Dirichlet condition at $\sigma = \pi$. Here the bosonic solution is

$$\varphi_{ND}(\tau, \sigma) = \varphi_0 + \frac{1}{\sqrt{\pi g}} \sum_{r \in \mathbb{Z} + \frac{1}{2}} \frac{1}{r} a_r e^{-r\tau} \cos(r\sigma). \tag{99}$$

Observe that the mode $a_0$ does not even act on $\mathcal{H}_{ND/DN}$. It has been twisted, $a_0 \to a_{\frac{1}{2}}$. Furthermore, $[L_0, a_{\frac{1}{2}}] = -\frac{1}{2} a_{\frac{1}{2}} \neq 0$, and thus this mode is no symmetry generator!

An algebraic, fully general, complementary argument for the breaking of the $U(1)$ symmetry uses the fact that the gluing automorphisms for Neumann, $J = \bar{J}|_{bdy}$, and Dirichlet, $J = -\bar{J}|_{bdy}$, are not compatible. By themselves, they each preserve a $U(1)$ symmetry, which we have already seen to be distinct. Hence, presence of mixed boundaries breaks the $U(1)$ symmetry, irrespective of the manifold in use.

# B  Modular Forms

Given the modular nome $q = e^{2\pi i \tau}$, the *Dedekind eta function* is

$$\eta(q) = q^{\frac{1}{24}} \prod_{n=1}^{\infty} (1 - q^n). \tag{100}$$

The *Jacobi theta functions* are

$$\vartheta_3(q) = \sum_{n \in \mathbb{Z}} q^{\frac{n^2}{2}} = q^{-\frac{1}{24}} \eta(q) \prod_{n=1}^{\infty} \left(1 + q^{n-\frac{1}{2}}\right)^2, \tag{101a}$$

$$\vartheta_2(q) = \sum_{n \in \mathbb{Z}} q^{\frac{1}{2}\left(n - \frac{1}{2}\right)^2} = 2 q^{\frac{1}{12}} \eta(q) \prod_{n=1}^{\infty} (1 + q^n)^2, \tag{101b}$$

$$\vartheta_4(q) = \sum_{n \in \mathbb{Z}} (-1)^n q^{\frac{n^2}{2}} = q^{-\frac{1}{24}} \eta(q) \prod_{n=1}^{\infty} \left(1 - q^{n-\frac{1}{2}}\right)^2, \tag{101c}$$

$$\vartheta_1(q) = i \sum_{n \in \mathbb{Z}} (-1)^n q^{\frac{1}{2}\left(n - \frac{1}{2}\right)^2} = \frac{1}{2} q^{\frac{1}{12}} \eta(q) \prod_{n=1}^{\infty} (1 - q^n)^2 = 0. \tag{101d}$$

The second equality in all these expressions follows from the *Jacobi triple product identity*

$$\prod_{n=1}^{\infty} (1 - q^n)(1 + q^{n-\frac{1}{2}} w)(1 + q^{n-\frac{1}{2}} w^{-1}) = \sum_{m \in \mathbb{Z}} q^{\frac{1}{2} m^2} w^m. \tag{102}$$

Modular transformations act on the modular parameter as follows

$$T: \quad \tau \to \tau + 1, \qquad S: \quad \tau \to -\frac{1}{\tau}. \tag{103}$$

The modular properties of the above modular functions are

$$\eta(\tau + 1) = e^{\frac{i\pi}{12}} \eta(\tau), \qquad \eta\left(-\frac{1}{\tau}\right) = \sqrt{-i\tau}\, \eta(\tau), \tag{104}$$

and

$$\vartheta_3(\tau + 1) = \vartheta_4(\tau), \qquad \vartheta_3\left(-\frac{1}{\tau}\right) = \sqrt{-i\tau}\vartheta_3(\tau), \tag{105a}$$

$$\vartheta_2(\tau + 1) = e^{\frac{i\pi}{12}} \vartheta_2(\tau), \qquad \vartheta_2\left(-\frac{1}{\tau}\right) = \sqrt{-i\tau}\vartheta_4(\tau), \tag{105b}$$

$$\vartheta_4(\tau + 1) = \vartheta_3(\tau), \qquad \vartheta_4\left(-\frac{1}{\tau}\right) = \sqrt{-i\tau}\vartheta_2(\tau). \tag{105c}$$

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
