# Peer review of "On the Boundary Conformal Field Theory Approach to Symmetry-Resolved Entanglement"

_SciPost Physics_

## Round 1 · Referee Report · Anonymous (Referee 1) · 2023-2-19

Strengths

  1. Application of well-developed tools of BCFT on the analysis of symmetry-resolved entanglement.
  2. Pedagogical, with reviews of previously obtained results.
  3. Provides a further resolution of the entanglement in the free-boson CFT in addition to the existing works.
  4. Potential for generalization to other interacting CFTs.

Weaknesses

  1. While the method seems to be more general, the authors consider only the free-boson CFT. It will be interesting to consider interacting models.
  2. It is unclear if this method can be applied for non-abelian symmetries.
  3. The new results constitute only 20% of the manuscript (Sec. 4 only), the rest being devoted to review of previously existing results, many of which have been known to the community since 1980-s.

Report

This work can lead to a deeper comprehension of entanglement properties of BCFTs. It will be particularly interesting to apply this to interacting CFTs .

Questions:

  1. Is the restriction to an abelian symmetry group a matter of convenience/ease of exposition or is it not clear how to formulate the problem at a theoretical level for non-abelian symmetries?

  2. The authors use the well-known boundary states of a free-boson. Would it be correct say that an explicit apriori knowledge of the boundary states is required to compute the symmetry-resolved entanglement properties?

---

## Round 1 · Referee Report · Anonymous (Referee 2) · 2023-4-24

Strengths

  1. Pedagogical and clear presentation of the computations.
  2. Interesting connection between entanglement properties and BCFT spectrum.

Weaknesses

  1. It does not add a significant contribution to the subject.
  2. The motivations, and the advantages of the method, are not particularly convincing.

Report

This work deals with the entanglement features of one interval in a CFT, relating them to the spectrum of a boundary CFT. The symmetry resolution is considered, and the paradigmatic example of the free boson is carefully analyzed.

I have just a few minor questions to the authors, related to the weakness I pointed out.

  • A key-point of the discussion is the need of regularization at the entangling points, as the "standard" factorization of the Hilbert space in tensor product does not hold for continuous quantum field. In particular, some small disks are inserted around the entangling points and boundary conditions are specified along them. Usually, one is interested in properties which are independent of the specific choice of the regularization procedure. In contrast, here most of the discussion is about the possibility of symmetry-resolving the entanglement, depending on the regularization considered. It is not clear to me if the conclusions made for the boson are physically relevant, or if they are just features/artifacts of the chosen regularization procedure.

  • One of the biggest advantage the authors suggest is the "bypass of the laborious computation of the charged moments". However, this sentence is not particularly clear to me... Firstly, because the charged moments are easily recovered in the formalism of the work, together with the other quantities. Secondly, for CFTs the charged moments are just two-point functions of primary (twist) fields (Ref.19 Goldstein, Sela) and its behavior within the subsystem size is rather transparent in that picture: the only missing piece of information is a proportionality constant, that is generically non-universal, and it seems related to the 'regularization-dependent' quantities computed in this work. Could you please elaborate a bit about this advantage of bypassing the charged moments?

  • While the generalization to other CFTs look straightforward, at least in principle, it is not clear if it is possible to generalize the method/ideas in the presence of a different subsystem geometry (say two intervals). Is it something you thought about?

  • validity: high
  • significance: good
  • originality: ok
  • clarity: high
  • formatting: excellent
  • grammar: excellent

Author:  Giuseppe Di Giulio  on 2023-05-09  [id 3654]

(in reply to Report 2 on 2023-04-24)

We are grateful to the referee for the comments, questions and improvements suggested. We have revised our manuscript and addressed these points. The account of the changes is given below, where we follow the order in the referee's original report.

  • We thank the referee for pointing out this aspect, which deserves more comments in our manuscript. The BCFT approach introduces a regularization of the CFT through the choice of boundary conditions. It is well-known that the leading order of the entanglement entropies and their symmetry resolution is universal and can be retrieved from the BCFT approach. On the other hand, the higher-order corrections in the UV cutoff are non-universal. This holds also for the results of the present manuscript. In particular, the equipartition at all orders in the UV cutoff expansion of the symmetry-resolved entropies into $U(1)$ charge sectors is a regularization-dependent finding. We have added a paragraph for stressing this fact in the final part of the third bullet point in section 5.1. In this paragraph, we comment on the motivations that lead us to mostly considered NN and DD boundary conditions. We also remark on the necessity of a comparison between the non-universal terms arising from the BCFT regularization and the ones in lattice models with the free boson CFT as the underlying continuum description.

  • We thank the referee for this comment. Indeed, the charged moments pose no problem in the case of $U(1)$. However, moving away from abelian groups, charged moments are mostly difficult to compute. In particular, no obvious twist field prescription as the one in Goldstein and Sela's work is available. For Wess-Zumino-Witten models charged moments can be evaluated after using the powerful tools of group theory for finite groups. However, for infinite-dimensional groups, e.g. the Virasoro group, representation theoretic tools such as Haar measures, do not carry over in an obvious way. It is thus desirable to have a complementary method to compute symmetry-resolved entropies in spite of these difficulties. Our method does indeed provide one such remedy, as has been shown for the Virasoro group recently (see Ref. [84] of the manuscript). We emphasize that our method is to be viewed as complementary to the charged moments, as examples exist where our method is difficult to implement, yet the charged moments are easily accessible. This is for instance the case in bottom-up holographic constructions. We have extended the discussion on the first bullet point of section 5.1 in our draft providing the clarifications mentioned here.

  • As discussed in Ref. [12,13] of the present manuscript, the subsystem geometry can be changed by modifying the function $W$ corresponding to the width of the annulus. However, the case of a subsystem given by multiple disjoint intervals cannot be treated within this approach since, after the regularization, the spacetime cannot be mapped to an annulus, but a more complicated geometry. To clarify this point, we have added a paragraph (the fourth, in the current version of the manuscript) in Sec.\,5.3, pointing out that the generalization of the BCFT approach to multi-interval geometries is an interesting future direction of research.

---

## Editorial Decision

resubmitted